# Towards a global understanding of vegetation–climate dynamics at multiple time scales

Nora Linscheid[1,2,*], Lina M. Estupinan–Suarez[1,3,*], Alexander Brenning[3,4], Nuno Carvalhais[1,5], Felix Cremer[3,6], Fabian Gans[1], Anja Rammig[2], Markus Reichstein[1,4,7], Carlos A. Sierra[1], and Miguel D. Mahecha[1,4,7]

[*]These authors contributed equally.
[1]Max Planck Institute for Biogeochemistry, Hans–Knoell–Str. 10, 07745 Jena, Germany.
[2]TUM School of Life Sciences Weihenstephan, Technical University of Munich, Hans–Carl–von–Carlowitz–Platz 2, 85354 Freising, Germany.
[3]Department of Geography, Friedrich Schiller University Jena, Loebdergraben 32, 07743 Jena, Germany.
[4]Michael Stifel Center Jena for Data–Driven & Simulation Science, Ernst–Abbe–Platz 2, 07743 Jena, Germany
[5]Departamento de Ciencias e Engenharia do Ambiente, DCEA, Faculdade de Ciencias e Tecnologia, FCT Universidade Nova de Lisboa, Caparica, Portugal
[6]Institute for Data Science, German Aerospace Center DLR, 07745 Jena, Germany
[7] German Centre for Integrative Biodiversity Research (iDiv), Deutscher Platz 5e, 04103 Leipzig, Germany.

**Correspondence:** Nora Linscheid (nlinsch@bgc–jena.mpg.de), Lina M. Estupinan-Suarez(lestup@bgc–jena.mpg.de)

**Abstract.** Climate variables carry signatures of variability at multiple time scales. How these modes of variability are reflected in the state of the terrestrial biosphere is still not quantified, nor discussed at the global scale. Here, we set out to gain a global understanding of the relevance of different modes of variability in vegetation greenness and its co–variability with climate. We used >30 years of remote sensing records of Normalized Difference Vegetation Index (NDVI) to characterize biosphere variability across time scales from sub–monthly oscillations to decadal trends using discrete Fourier decomposition. Climate data of air temperature ($T_{air}$) and precipitation (Prec) were used to characterize atmosphere–biosphere co–variability at each time scale.

Our results show that short–term (intra–annual) and longer–term (inter–annual and longer) modes of variability make regionally highly important contributions to NDVI variability: Short–term oscillations focus in the tropics where they shape 27% of NDVI variability. Longer–term oscillations shape 9% of NDVI variability, dominantly in semi–arid shrublands. Assessing dominant time scales of vegetation–climate co–variation, a natural surface classification emerges which captures patterns not represented by conventional classifications, especially in the tropics. Finally, we find that correlations between variables can differ and even invert signs across time scales. For southern Africa for example, correlation between NDVI and $T_{air}$ is positive for the seasonal signal, but negative for short–term and longer–term oscillations, indicating that both short and long–term temperature anomalies can induce stress on vegetation dynamics. Such contrasting correlations between time scales exist for 15% of vegetated area for NDVI with $T_{air}$, and 27% with Prec, indicating global relevance of scale–specific climate sensitivities.

Our analysis provides a detailed picture of vegetation–climate co–variability globally, characterizing ecosystems by their intrinsic modes of temporal variability. We find that (i) correlations of NDVI with climate can differ between scales, (ii) non–dominant sub–signals in climate variables may dominate the biospheric response, and (iii) possible links may exist between

short–term and longer–term scales. These heterogeneous ecosystem responses on different time scales may depend on climate zone and vegetation type, and are to date not well understood, nor always correspond to transitions in dominant vegetation types. These scale dependencies can be a benchmark for vegetation model evaluation and for comparing remote sensing products.

## 1   Introduction

Ecosystems and climate interact on multiple spatial and temporal scales. For example, the main driver of photosynthesis during the daily cycle typically is light availability, assuming no other resource limitation. At annual time scales, temperature can limit growth and development during certain phases of the year, particularly in the extratropics. While climate variability is traditionally very well characterized across time scales (e. g. Viles (2003); Cao et al. (2012); Bala et al. (2010); Hannachi et al. (2017)), it is less well known how the biosphere responds to variations in climate on different scales. Understanding the

implications of such time–scale dependencies of climate–vegetation interactions is challenging due to the variety of interwoven processes. These dependencies range from short–term climate extremes and biotic stress (e. g. insect outbreaks) to seasonal dynamics in climate–driven phenology and long–term dynamics that can again either reflect intrinsic ecosystem dynamics (e. g. vegetation successional dynamics) or climate–change or land–use induced process alterations. Investigating vegetation–climate dynamics globally across multiple time scales requires long–term observation on relevant vegetation dynamics and

climate variables in combination with a method to separate ecosystem variability at different time scales.

    The assessment of ecosystem variability e. g. in responses to climate at the global scale has only become feasible in the last decades. Long–term Earth observations (EOs) are now allowing us to assess ecosystem states consistently over more than 30 years. Vegetation indices such as the Normalized Difference Vegetation Index (NDVI) have often been interpreted as proxies for vegetation activity (Zeng et al., 2013; De Keersmaecker et al., 2015; Hawinkel et al., 2015; Kogan and Guo, 2017; Pan et al.,

2018), despite well–known limitations of only reflecting vegetation greenness. While novel EOs may be more closely related to actual rates of photosynthesis (e. g. Solar Induced Fluorescence, SIF (Guanter et al., 2007)), NDVI from the Advanced Very High Resolution Radiometer (AVHRR) has the advantage of offering the longest updated records of vegetation remote sensing data every 15 days. In tandem with climate time series from the same period, this record provides a solid basis to globally assess biosphere–atmosphere interactions across time scales ranging from weeks to decades.

Temporal biosphere dynamics carry the imprint of different drivers across time scales, yet EOs can only record one integrated signal over time. This signal reflects a mixture of processes acting on different scales, which cannot be observed independently (Mahecha et al., 2007; Defriez and Reuman, 2017; Pan et al., 2018). Therefore, short–term and long–term processes can be obscured by the dominant influence of the annual cycle (Braswell et al., 2005; Mahecha et al., 2010c). In order to study relevant ecosystem–climate interactions across temporal scales, information contained for each time scale thus first needs to be extracted

from this integrated signal. Time–series decomposition allows to extract different frequencies such as annual, intra–annual and inter–annual oscillations from vegetation and climate time series. Such approaches have proven useful e. g. to characterize at what scales vegetation responses are dampened or amplified in comparison with their climate forcing (Stoy et al., 2009),

how ecosystem variability is confined by hydrometeorological variability (Pappas et al., 2017), what scales of variability need to be considered to relate forcing variables and vegetation state comprehensively (Katul et al., 2001; Braswell et al., 2005), or to remove confounding effects from processes acting on longer time scales than the process in question (Mahecha et al., 2010b). However, to date most studies employing time–series decomposition to study vegetation dynamics have focused on disentangling time scales from minutes to few years based on flux data (Stoy et al., 2009; Katul et al., 2001; Mahecha et al., 2007, 2010c). Studies investigating long–term vegetation records by time–series decomposition do exist, but focus only on a specific region (Martínez and Gilabert, 2009; Canisius et al., 2007; Hawinkel et al., 2015) or do not provide co–interpretation with climate signals (Pan et al., 2018). Earth observation time series of vegetation and climate covering more than 30 years now allow us to characterize the time–scale resolved variability in the biosphere and its relation to climate globally across several decades. Additionally, the global coverage of these records allows one to attain a broader understanding in climate space and across vegetation types, which to date is equally lacking.

In this study, we set out to gain a global understanding of the relevance of the different modes of variability in vegetation greenness and its co–variability with climate at time scales from sub–monthly oscillations to long–term trends. These time–scale–specific vegetation–climate co–oscillations are expected to serve as reference benchmark for comparing remote sensing products and terrestrial biosphere models. Specifically, we aim to (i) characterize variability of biosphere and climate time series explicitly on multiple time scales, (ii) understand spatial patterns of this scale–resolved variability and co–variability globally, (iii) assess whether characteristic time–scale specific dynamics in biosphere and climate relate to established climate classifications or land cover, and (vi) assess differences in correlations of biosphere with climate on short–term, seasonal, and longer–term time scales.

## 2 Methods

The code to produce all primary figures is made available as supplementary notebook.

### 2.1 Data

A global gridded dataset of NDVI AVHRR was retrieved from the Global Inventory Monitoring and Modeling System (GIMMS, Pinzon and Tucker, 2014) at 15–daily temporal and 0.083° spatial resolutions (GIMMS NDVI v3.1). Original data was aggregated to 0.5° by taking the mean of corresponding 0.083° pixels. Corresponding records of air temperature ($T_{air}$), from the European Centre for Medium–Range Weather Forecasts (ERA Interim v4, Dee et al., 2011) and precipitation (Prec) from the Multi–Source Weighted–Ensemble Precipitation (MSWEP, Beck et al., 2019) were aggregated to match temporal resolution by summation (Prec) or averaging ($T_{air}$). Spatial resolution of $T_{air}$ was preserved (0.5°), while MSWEP values were averaged for spatial resampling (0.083° to 0.5°). Spatial and temporal resolution were fixed based on the coarsest resolution among the input datasets to ensure conservative results. The time period considered was from 1 January 1982 to 31 December 2015.

## 2.2 Pre–processing

Gaps in NDVI time series were filled with values from the mean seasonal cycle computed separately for each grid cell. Missing values were mostly present at high Northern latitudes (Sup. Fig. S1). Each time series (for each pixel) was normalized to zero mean and unit variance prior to performing Fast Fourier transformation (FFT). For further analysis, the gap filled data was discarded. Normalization, gap filling and FFT were performed in the Earth System Data Lab (https://www. earthsystemdata-lab.net/, Mahecha et al., 2019), using the implementation based on the programming language Julia. Analyses were performed on Lat–Lon grid due to software and data considerations. In all spatial analyses on Lat–Lon grid, the difference in size of grid cells between high latitudes and the equator was accounted for through weighting values by grid cell size. Similarly, in all analyses that involved sampling of data points, the sampling frequency was weighted by grid cell size.

## 2.3 Time series decomposition

All pixel time series were first detrended using a linear model. We then used discrete FFT to decompose the detrended time series into underlying harmonic functions at different frequencies (Brockwell and Davis, 2006). The resulting Fourier spectra (Sup. Fig. S2) were reconstructed by inverse FFT into binned scale–specific sub–signals for short–term, seasonal and longer–term oscillations: The seasonal signal was reconstructed from the Fourier spectrum at periods of 0.9–1.1 years, plus semi–annual and 4–monthly harmonics at 0.5–yr and 0.33–yr periods. The short–term signal was reconstructed from the Fourier spectrum of all periods <0.9 year, except the two seasonal harmonics, representing inter–annual oscillations that are not directly linked to periods of seasonality. The longer–term signal was reconstructed from all remaining periods >1.1 year, representing inter–annual and longer time scales. The sub-signal binning was centered on the definition of the seasonal/annual bin similarly to Mahecha et al. (2010a) and Fürst (2009). The bin ranges were slightly adapted due to the FFT approach, which yields signals of different frequencies compared to the approach chosen by Mahecha et al. (2010a). To identify emerging features occurring at different latitudinal bands, mean values weighted by pixel area were calculated in the tropics (23.5°N to 23.5°S), extratropics (above 23.5°N and below 23.5°S), and globally.

## 2.4 Variance per time scale and co–oscillation regimes

For each time scale–specific signal, we calculated the proportion of variance of the original signal explained for each variable per grid cell. Each pixel of the global land surface was then classified into oscillation regimes depending on which scale explained the largest amount of variance in each variable (abbreviations: S – short–term, A – seasonal, L – longer–term, T – trend). For example, if the variance was dominated by the seasonal sub–signal in NDVI and $T_{air}$, and by the short–term scale in Prec, this pixel would be classified as AAS (in order NDVI, $T_{air}$, Prec). Theoretically, the superimposition yields 64 ($4^3$) possible combinations, of which only 26 occurred. For simplicity, our analysis was focused on the 11 most abundant oscillation regimes (99.7% of pixels).

In order to complement static/traditional classifications, we compared our oscillation regimes with the Global Land Cover Map Project coordinated by the Joint Research Center (GLC2000, Bartholomé and Belward, 2005), and climate zones from the

updated Köppen–Geiger global classification (Kottek et al., 2006, see Sup. Fig. S1). Only those pixels that contained data from all three data streams (Köppen–Geiger classes A–D, GLC2000, and our oscillation regimes) were considered in this analysis. Non–vegetated and non–natural areas as defined by GLC2000 were disregarded for this analysis and onward (Sup. Table S1). The final land surface assessed was 75,871,486 km$^2$, corresponding to 70% of vegetated GLC2000 area (Sup. Fig. S1). For the same area, we calculated the V–measure (V), a spatial association index based on homogeneity and complementarity criteria proposed specifically for thematic map comparison (Nowosad and Stepinski, 2018). The index ranges from 0 to 1, with 1 being a perfect association, and was used to provide an overall comparison between the co–oscillation regime map with Köppen–Geiger and GLC2000 maps.

To assess the influence of gap–filling performed in the original GIMMS NDVI data due to influence of cloud cover or snow, we excluded time points that were retrieved by splines or mean seasonal cycle due to lack of direct observation in NDVI (Pinzon and Tucker, 2014) at five different quality flag thresholds in our classification of oscillation regimes. Quality flags were aggregated from 0.083° to 0.5° by calculating the fraction of direct observations per 0.5° pixel at each time step. Subsequently, the dominant classification was repeated, excluding time steps with less than 30%, 50%, 70%, 90% and 95% direct observations for each grid cell. Furthermore, we repeated the time series decomposition method for NDVI and the Enhanced Vegetation Index (EVI) from the Moderate Resolution Imaging Spectroradiometer MODIS. The vegetation indices product MOD13C1.006 is provided by NASA EOSDIS LP DAAC at 0.05°. Data was aggregated spatially by averaging valid pixels to 0.5° for the overlapping period with GIMMS NDVI (2001–2015). A comparison of the dominant oscillations regimes between products was carried out at pixel basis.

## 2.5 Correlations between variables at each time scale

We correlated time scale–specific sub–signals of NDVI, $T_{air}$ and Prec using Pearson's correlation coefficient, Spearman correlation, and partial correlation. For this analysis, all time points with NDVI<0.2 were masked in order to consider only data points corresponding to active vegetation (Sup. Fig. S1). NDVI was lagged one time step (15 days) behind Prec in order to allow response time of vegetation to changes in water availability. Due to the 15–daily temporal resolution of the data, a response time of up to 15 days is intrinsically included in our analyses. Each time lag is therefore an additional 15 days, and shorter responses cannot be assessed. We compared six different lags (from 15 to 90 days, Sup. Fig. S3). When correlating NDVI and precipitation instantaneously, we found almost exclusively negative correlations for the short–term scale. A lag of one time step was sufficient to arrive at expected positive correlations between NDVI and precipitation, while increasing the lag time did not substantially improve or alter the results. We thus chose to globally use a lag of one time step (representing 15–30 days response time) between precipitation and NDVI across all scales. Globally, temperature appeared to be most strongly correlated to NDVI instantaneously (not lagged), thus no time lag was introduced between air temperature and NDVI. Recent studies assessing time lags and memory effects between vegetation and climate also indicate that time lags of around one month generally carry most explanatory power for predicting vegetation dynamics (Krich et al., 2019; Kraft et al., 2019; Papagiannopoulou et al., 2017). Correlation of NDVI–$T_{air}$ and NDVI–Prec were binned into five quantiles and presented in a

bivariate color map (Teuling et al., 2011). In addition, we compared differences in the sign of correlation (+/– or –/+) between seasonal and longer–term oscillations to detect areas where the correlation was inverted between scales.

## 2.6 Assessment of land cover change on time series decomposition

We assessed whether land cover change over the 30–year time period influenced our results by extracting pixels with substantial land cover change as determined by Song et al. (2018). While linear trends were removed from the time series before decomposition, changes in amplitude or piecewise linear and non–linear trends may have an impact on our analyses. First, we aggregated original 0.05° data to match our 0.5° spatial resolution by averaging. We then determined 0.5° pixels with >25% gain or loss of trees, short vegetation, or bare ground, and assessed whether the observed changes in land cover (Song et al., 2018) were reflected in the NDVI time series to a degree that substantially affected the classification of dominant oscillation regimes.

## 2.7 Comparison of Fourier Transform with Empirical Mode Decomposition

While the FFT approach is the most classical time–series decomposition technique, there are more data adaptive alternatives available (Huang et al., 1998; Ghil, 2002; Paluš and Novotná, 2008). In order to understand whether different methods would lead to different insights, we compared the employed FFT approach with the more data–adaptive Empirical Mode Decomposition (EMD). EMD repeatedly extracts sub–signals (intrinsic mode functions, IMFs) from the time–series by interpolating a spline between local minima and maxima until the residuals converge to approximately constant values (Huang et al., 1998). We used an ensemble–based modification of the EMD algorithm, the complete ensemble empirical mode decomposition with adaptive noise (CEEMDAN, Colominas et al., 2014; Torres et al., 2011) and a frequency binning approach to obtain frequency bands comparable to the ones chosen for FFT. In contrast to the regular EMD, CEEMDAN employs an ensemble approach in which noise is added to the data before decomposition and ensemble averages for each IMF are returned, so that a more robust end result is obtained (Colominas et al., 2014; Torres et al., 2011). Briefly, in CEEMDAN each IMF is computed as the mean of an ensemble of IMFs retrieved from noisy data copies. This IMF is subtracted from the original signal, and the residual signal used as input for retrieving the next IMF (Colominas et al., 2014; Torres et al., 2011). As such, CEEMDAN is less prone to mode mixing than EMD while still fulfilling the completeness property of EMD (i. e. the sum of all IMFs equals the original signal). As IMFs resulting from EMD do not have a fixed frequency assigned, we then associated each IMF with a time scale by measuring the distance between all local maxima and minima as a proxy for the dominating wavelength of the signal. Distances between each two maxima or minima were classified as short–term, seasonal or longer–term depending on their length. The IMF was then categorized by the majority distance category and added into the respective time scale bin. For example, if an IMF contained 25 seasonal cycles and 5 short–term cycles, it was classified as seasonal and added to the seasonal signal bin. IMFs in each bin were combined by summation.

## 3 Results

### 3.1 Time series variance across time scales

Assessing the contribution of each time scale sub–signal to the signal variance at each grid cell, we find that for NDVI most of the temporal variability is expectedly captured by the seasonal cycle (71% of the global variance), especially above the Tropic of Cancer (23.5°N) (Fig. 1, Sup. Table S2). Short–term oscillations contribute dominantly in parts of tropical America and Southeast Asia, while longer–term components are mainly observed in Australia, South Africa, parts of Argentina, and northern Mexico. Specifically, short–term and longer–term signals together contribute with 27% of total NDVI variance globally, and with 38% in the equatorial region (23.5°N to 23.5°S).

Similarly, $T_{air}$ is strongly dominated by seasonal oscillations in the extratropics above/below 23.5° N/S (94% and 90%, respectively, Sup. Table S2) as would be expected. Even in the tropics, short–term and longer–term components contribute with only 30% of variance (and 11%, 4% of global variance, respectively, Sup. Table S2). In contrast, short–term oscillations dominate global precipitation variance before the seasonal cycle (52% and 41% of global variance each, Sup. Table S2). An East–West gradient of precipitation over Eurasia stands out, changing from predominantly short–term to predominantly seasonal signal variance. In the tropics, a similar contribution from both oscillations is found (42% and 41%, respectively, Sup. Table S2). Linear trends removed before FFT decomposition had a minor influence on overall variance (Fig. 1). In summary, short–term and longer–term signals show substantial, regionally focused contributions to signal variance. These regions differ between variables, suggesting complex patterns of temporal interaction.

### 3.2 Classification of co–oscillations regimes

Given the contrasting, spatially heterogeneous patterns observed in different variables in Fig. 1, we investigated how scale–specific oscillations of biosphere and climate co–occur globally. We combined the dominant scale of variability for each variable in each grid cell (Sup. Fig. S4) and found that 84.5% of the assessed area is dominated by seasonal oscillations of NDVI, 9% by short–term oscillations in NDVI, and 6.5% longer–term oscillations in NDVI (0.03% captured by the trend). Combining the maps for all three variables into a map of co–dominant "oscillation regimes" (Fig. 2, Sup. Table S3), we find that seasonal NDVI regimes co–occur predominantly with seasonal $T_{air}$, and seasonal or short–term Prec regimes (blue regions). Dominant seasonal cycles of NDVI and $T_{air}$, as well as fast oscillation regimes in Prec, are expected over large parts of the globe, which is reflected by the large extent of the AAS and AAA classes in this analysis. Beyond this expected, solar cycle induced behaviour, a number of differentiated oscillation classes stands out: Short–term NDVI oscillations occur mainly in the South American and Asian tropics, in a multitude of combinations with predominantly seasonal or short–term $T_{air}$ and Prec (light green, red and light red regions). Longer–term oscillation regimes of NDVI co–occur with seasonal $T_{air}$ and short–term Prec regimes (dark green regions) around the west side of South Africa, east side of southern South America, and Australia. Interestingly, the dominant scales in climatic variables are not always associated with similar dominant regimes in NDVI dynamics, suggesting complex or additional driving mechanisms in these heterogeneous regions. In fact, even in areas where temperature or precipitation has a seasonal cycle, NDVI can be dominated by short–term or longer–term oscillations:

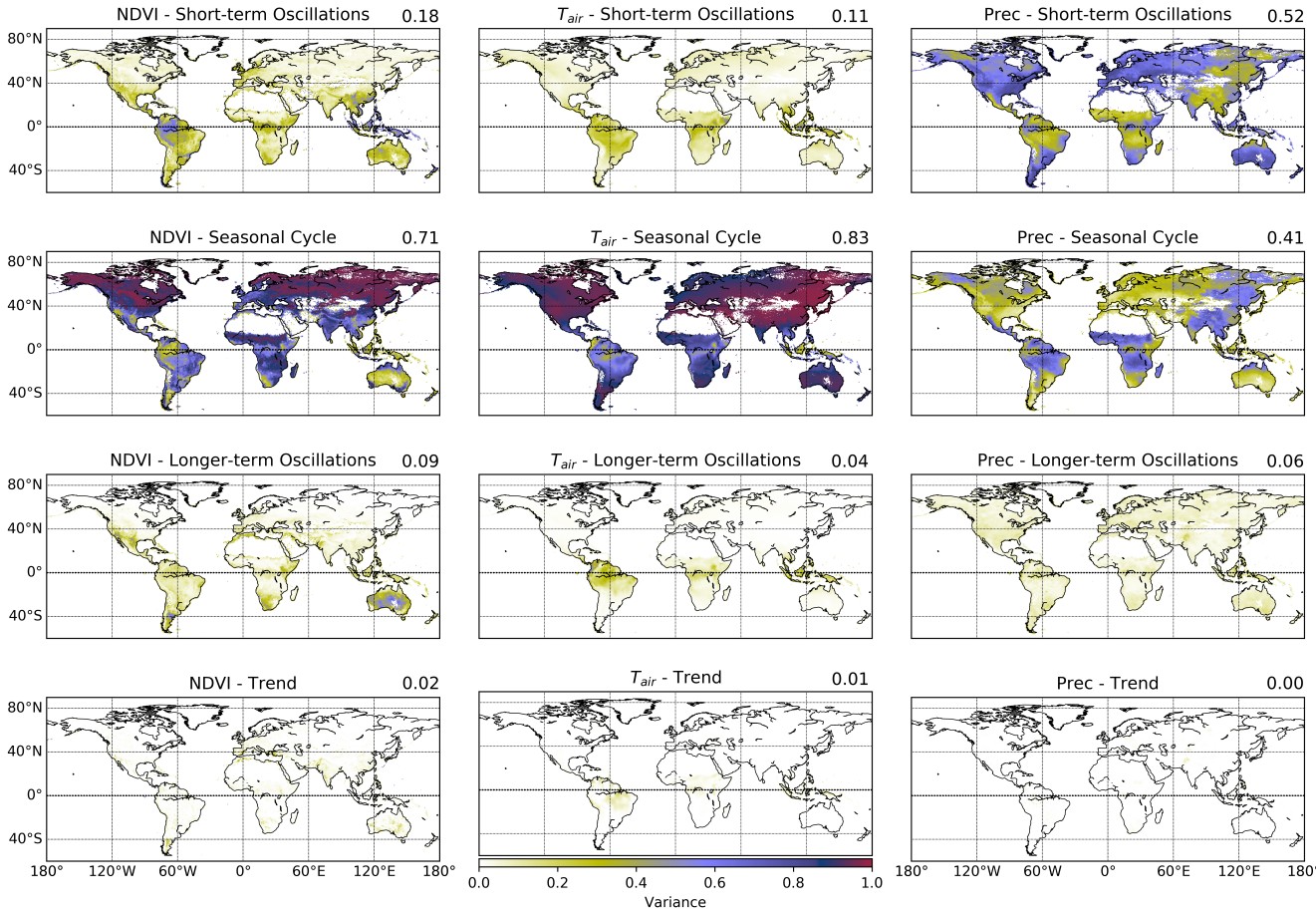

**Figure 1. Global distribution of time–scale specific variance (relative spectral powers) of NDVI, temperature ($T_{air}$), and precipitation (Prec).** Normalized time series of NDVI, $T_{air}$ and Prec (columns) were decomposed by Fast Fourier transformation and reconstructed into short–term (intra–annual), seasonal(annual), and longer-term (inter–annual) components (rows). Relative contribution of each scale–specific signal to overall variance was determined at each grid cell. Globally, most variance of NDVI and $T_{air}$ is contained in the seasonal component (red colors), while Prec shows a high contribution of variance from the short–term component. The semi–annual cycle is included in the seasonal band. Upper right corner values show the percentage of overall variance explained by each time scale.

More than 90% of the area with short–term NDVI regimes exhibits predominantly seasonal $T_{air}$, of which 36% also show predominantly seasonal Prec (SAA), and 55% predominantly short–term Prec (SAS, Sup. Table S3). All areas where NDVI is predominantly longer–term are classified as seasonal $T_{air}$ and short–term Prec regimes (LAS, Sup. Table S3).

To account for the influence of clouds and snow cover in the GIMMS NDVI record, especially in the tropics and northern regions, we excluded time points where pixels contained a high proportion of gap–filled values. We found that overall less than 1.5% of pixels changed their dominant oscillation class when only pixels with more than 0.7 direct observation fraction were considered. Even when the highest quality threshold was applied (0.95 direct observation fraction), only 2.6% of pixels changed dominant oscillation class (Sup Fig. S3). Short–term pixels were the most affected by changes in dominant oscillation (12.9% and 20.8% for 0.7 and 0.95 direct observations threshold respectively), while seasonal pixels showed the highest fraction of gap–filling overall (Sup. Fig. S5). As a further validation, we found very similar results when repeating the time series decomposition and the dominant oscillation regime classification based on EVI and NDVI from MODIS (Didan et al., 2015; Huete, 1997; Huete et al., 2002) for the years 2001–2015 (Sup. Fig. S6).

We investigated to what extent our classification into "oscillation regimes" shows patterns of temporal vegetation–climate relations that are not represented by conventional "static" classifications of the land surface. To determine overlap and differences between the classification of temporal vegetation–climate co–oscillations with static classifications of land cover (GLC2000) and Köppen–Geiger climate classes we assessed their spatial association by the V–measure (Nowosad and Stepinski, 2018). The V–measures of co–oscillation regimes with Köppen–Geiger and GLC2000 were V = 0.17 and V = 0.11, respectively, indicating weak association to both static classifications. Hence, our classification contains information largely complementary to the compared climate and land cover classfications. Yet we observed a slightly stronger association with Köppen–Geiger than with GLC2000, also when comparing homogeneity and complementarity (Sup. Table S4). Comparing the three classifications among each other, we find that dominant temporal patterns in NDVI can be linked to certain land cover types such as shrubs and broadleaf forest: Sankey diagrams (Fig. 2b+c) display which proportion of land surface is commonly classified across different class combinations in the three data layers of co–oscillation regime, GLC2000, and Köppen–Geiger for evergreen broadleaf forest (EBF, Fig. 2b) and areas dominated by longer–term NDVI (Fig. 2c). We find that EBF is the most diverse among land cover classes in terms of our temporal classification, with 35% dominated by short–term NDVI oscillation (Fig. 2b). In contrast, more than 95% of deciduous and evergreen needleleaf forests (DNF, ENF) and deciduous broadleaf forests (DBF) are dominated by seasonal NDVI regimes (Sup. Table S3). We further find a strong association of longer–term NDVI regimes with shrubs (21% of the area dominated by longer–term NDVI), herbaceous (26%) and sparse shrubs/herbaceous (49%) land cover types in arid regions (Fig. 2c, overall 93% of LAS area coincides with Köppen–Geiger class B). Thus, differences within and among land cover and climate types exist when assessing temporal co–oscillations of vegetation and climate.

### 3.3 Assessment of land cover change on time series decomposition

In the above analyses we did not aim to explicitly detect the effect of land cover or land use change (LCLUC), but nevertheless LCLUC could have an influence on our NDVI classification (Fig. 2). We assessed whether changes in vegetation cover over the 30–year period severely affected our classification by inspecting pixels with >25% change in fraction of trees, short

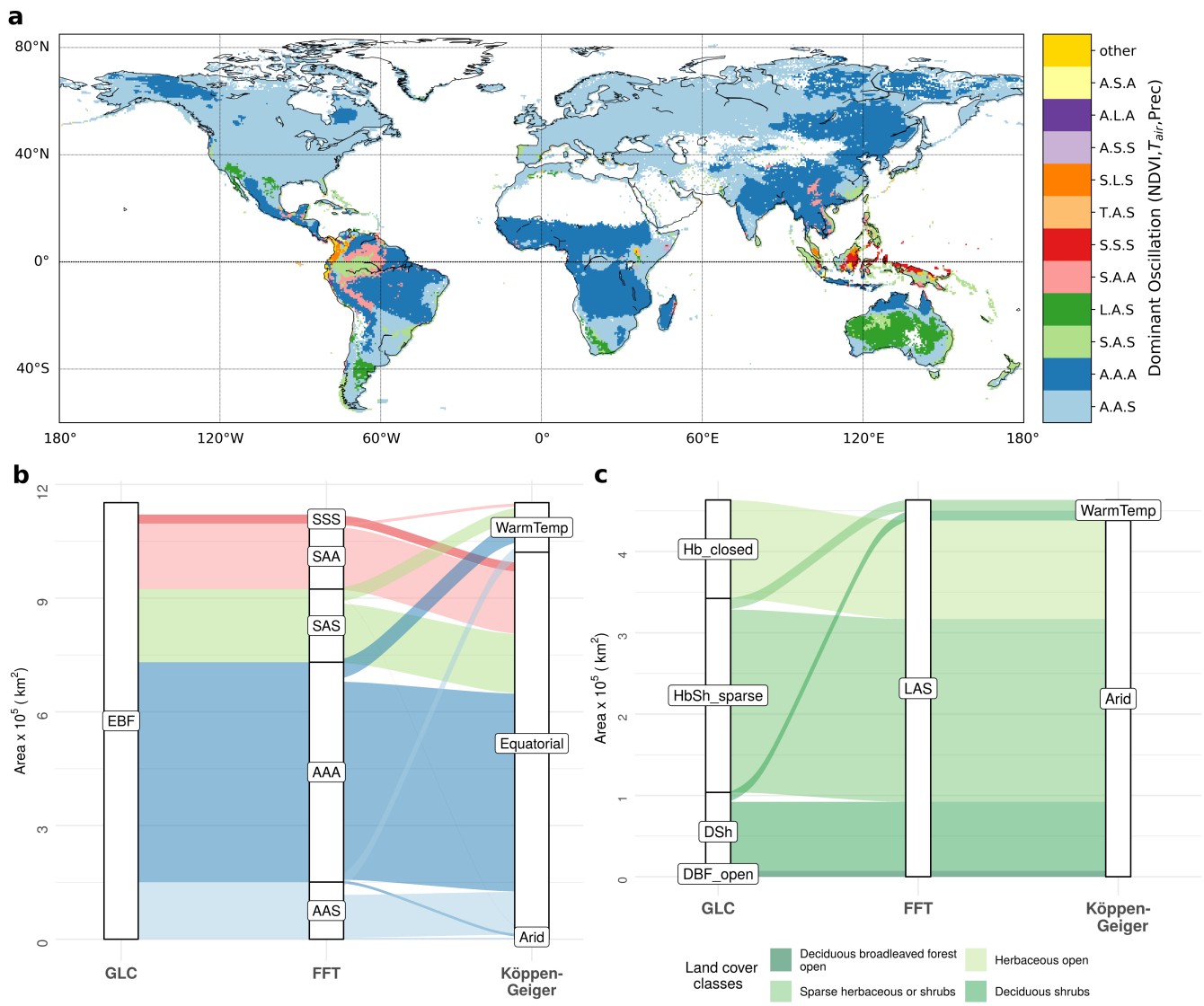

**Figure 2. Classification of land surface by dominant scale of variability in NDVI and climate, and its relation to land cover and mean climate. a.** Dominant scale of variability was determined for NDVI, $T_{air}$ and Prec separately for each grid cell and summarized as unique combinations between variables (S – short–term, A – seasonal, L – longer–term, T – trend, listed in order NDVI, $T_{air}$, Prec). Only the 11 most common classes are shown. The semi-annual cycle is included in the seasonal band. **b+c.** Sankey diagrams (river plots) showing associations of pixels for (b) Evergreen broadleaf forest (EBF) and (c) regions of dominant long–term oscillations in NDVI (LAS class) to oscillation regime (FFT), land cover class (GLC2000) and Köppen–Geiger (KG) climate class. The width of the ribbons is proportional to the area that is commonly classified into the corresponding GLC2000, KG or oscillation classes. DBF: Deciduous broadleaf forest, Hb_closed: closed herbaceous land cover, DSh: deciduous shrublands, HbSh_sparse: sparse herbaceous and shrub vegetation, Equatorial: KG class A, Arid: KG class B, WarmTemp: KG class C.

245 vegetation, or bare ground according to (Song et al., 2018). Notably, very few of such pixels showed marked signs of land cover change reflected in NDVI time series at all, which is likely due to the coarse spatial resolution of the data used in this study as compared to previous studies focused on detecting LCLUC (Song et al., 2018; Fensholt et al., 2015): at half–degree resolution, most pixels represent mixed signals which obscure most of the details that would allow for detecting land cover changes. In those pixels where we did see a clear progression in NDVI over time, the method did adequately capture this

progression, e. g. by correctly reflecting an increasing amplitude of the seasonal cycle and/or shifting baseline (Sup. Fig. S7). However, most of such pixels with pronounced positive or negative NDVI progression located in agricultural areas or areas of urbanization, which had *a priori* been excluded from downstream analyses. Overall, the change of vegetation over time did not have a widespread influence on the classification of dominant scale and oscillation regimes at the given spatial resolution.

### 3.4 Correlations of NDVI with climate on multiple scales

To inspect relationships of vegetation with climate at multiple time scales, we correlated NDVI with $T_{air}$ and Prec at each pixel for each time scale (Fig. 3). We found different correlation patterns depending on the time scale: while all possible combinations of correlation between NDVI and $T_{air}$ or Prec exist at the seasonal scale, short–term and longer–term scales show predominantly $T_{air}$+/Prec– (+/–) or $T_{air}$–/Prec+ (–/+) relationships. On the seasonal scale, NDVI correlates positively with $T_{air}$ and Prec above 40°N, whereas in the other latitudes all possible relations are observed. Especially South America

shows a highly diverse pattern of correlations. Differences exists across the tropics, where South America and South–East Asia display mainly negative correlation with Prec, whereas African tropics display positive correlation with Prec. Semi–arid regions show negative correlations with $T_{air}$ as would be expected. While some of the patters are known, this correlation of decomposed oscillations reveals a more differentiated picture of ecosystem variability in comparison with the undecomposed data (Sup. Fig. S8). Notably, correlations on short– and longer–term scales partially show opposite signs compared to the

seasonal scale, e. g. in South America, South Africa and Central America. Repeating the analysis with Spearman correlation and partial correlation returned similar results (Sup. Fig. S9–10). Due to the known saturation effects of NDVI against plant productivity over areas of dense biomass, we repeated the analysis with MODIS EVI. We found overall similar results across time scales, but correlations with $T_{air}$ turned from negative to positive in parts of Central and South America, as well as India (Sup. Fig. S11), indicating that NDVI saturation may affect the results obtained from GIMMS long–term records in some areas.

We again compared the observed patterns with vegetation types, to understand how different ecosystems react at different time scales, and found that different land cover classes showed distinct correlation patterns (Fig. 3b+c). Broadleaf evergreen forest shows the most diverse correlations on seasonal scale (Fig. 3b). For short–term oscillations, the strongest correlations were found in semi–arid shrublands and savannas, which spatially coincide with patterns observed in the longer–term: for longer–term oscillations, the strong correlation Prec+ and $T_{air}$– was again related primarily to shrublands and savannas (Fig. 3a

blue areas, Fig. 3c). We also observed a widespread positive longer–term correlation of NDVI with $T_{air}$ in the northern latitudes.

Comparing with static classifications, we found that Köppen–Geiger climate classes had the most prominent differentiating effect for correlation patterns, and different climate classes occupy distinct patterns in this "correlation space" across scales

(Sup. Fig. S12). Different land cover types generally show similar correlations within one climate zone, but exceptions exist (Sup. Fig. S13). Most prominently, EBF shows the most heterogeneous, spatially varying correlations on seasonal scale. All land cover types show a confined correlation pattern of mainly $T_{air}$+/Prec– or $T_{air}$–/Prec+ at the longer–term scale (Fig. 3c), which is further differentiated by climate zone (Sup. Fig. S14).

Assessing correlations across the different time scales, we find that the majority of northern temperate regions (Köppen class D) is positively correlated with $T_{air}$ on all time scales, but correlation with Prec varies (zero for short–term and longer–term, seasonal scale: generally negative on the coast, positive in the interior continent). The equatorial region, South America, Africa, and Southeast Asia exhibit different correlation patterns with climate despite similar land cover types (tropical forest). In some regions, opposing correlations can be observed across time scales (Fig. 4a). For example, correlation of NDVI with $T_{air}$ in South Africa varies from negative on the short–term scale, over positive on seasonal scale, back to negative on longer–term scale. As another example, on the east coast of Australia, NDVI has low correlation with precipitation on the seasonal scale but high in the longer–term. Assessing this globally, correlations between NDVI and $T_{air}$ show inverted signs between seasonal and longer–term scales in 15.4% of vegetated land surface area (Fig. 4a+c). The same is true for NDVI and Prec in 27.3% of vegetated land surface area (Fig. 4b+c).

In summary, we find that correlations between NDVI and climate variables can change strongly between time scales. Semi–arid ecosystems show most prominent short–term and longer–term signatures, while tropical rainforest show the most diverse relationships between variables. These patterns point to complex ecosystem responses to climate at different time scales, indicating that scale–specific ecosystem characterization is necessary to fully understand their temporal dynamics.

### 3.5 Comparison of Fast Fourier Transformation with Empirical Mode Decomposition

FFT decomposes a signal in the frequency domain under the assumption that the underlying signals are sinusoidal, time–invariant, and additive (Brockwell and Davis, 2006). Although resulting power spectra and frequency–invariant modes of oscillation are conveniently interpretable, not all ecological processes can be expected to follow regular periodic and additive oscillatory patterns approximated by sine and cosine waves over time. We chose FFT decomposition due to its superior computational speed and stable global applicability, i. e. its ability to return homogeneous spatiotemporal patterns in our analysis. To ensure that the above limitations did not confound our results, we compared the FFT approach to the data–adaptive empirical mode decomposition (EMD), which could be expected to be better suited for exploring instationary ecological processes over time. In a test case over Europe, we found that our binning approach resulted in comparable results for the two methods, both in terms of spatial and temporal behaviour of the signals (Sup. Fig. S15–18). CEEMDAN generally attributed slightly less signal variance to the short–term and slightly more to the seasonal cycle for both $T_{air}$ and NDVI and generally showed less modulation in the longer–term signals. Nevertheless, overall results were remarkably comparable. However, because CEEMDAN is a data adaptive method a higher spatial heterogeneity and spatially varying sensitivity to the noise parameter were observed, which currently constrains a global implementation of the analysis.

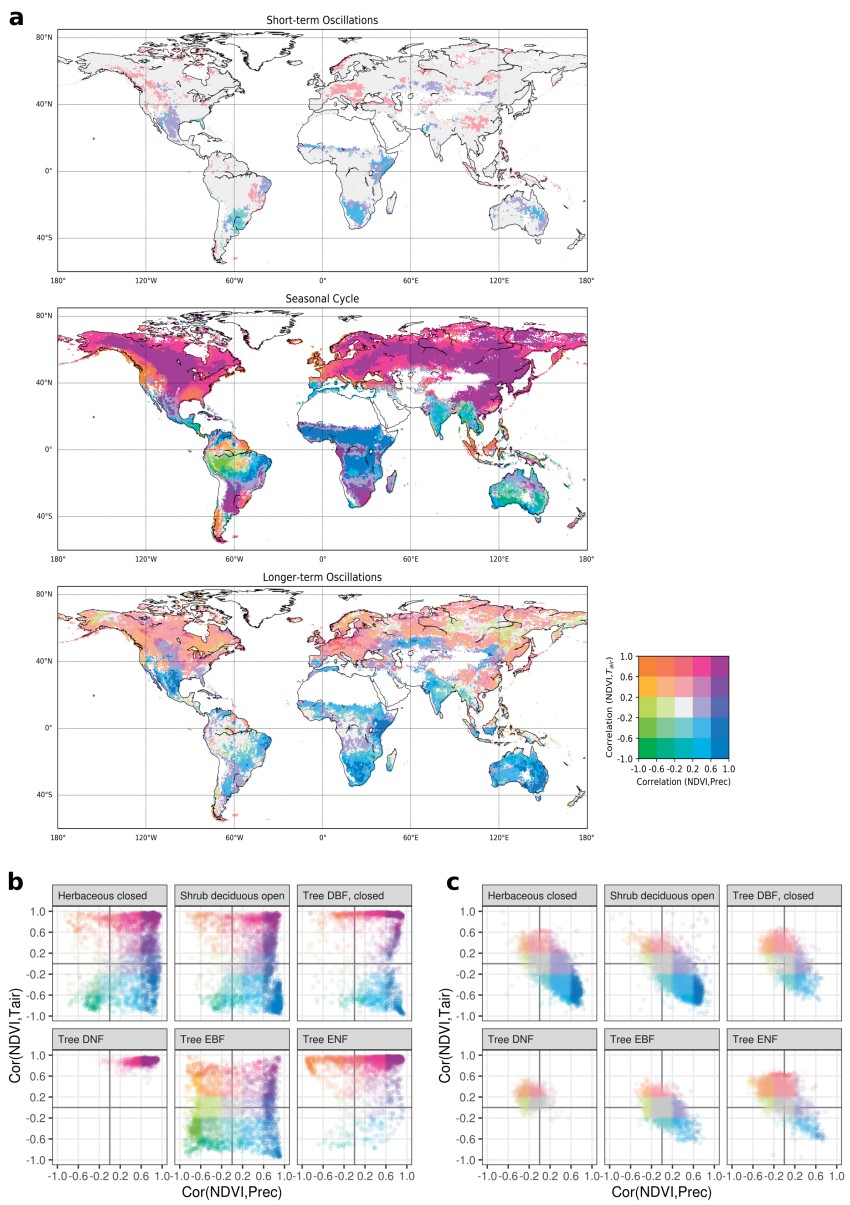

**Figure 3.** . **Global distribution of time scale–specific correlation of NDVI with air temperature ($T_{air}$) and precipitation (Prec). a.** Correlation of NDVI with $T_{air}$ and NDVI with Prec were calculated between decomposed signals at each grid cell. NDVI was lagged one time step (15 days) behind precipitation to allow response time, $T_{air}$ was correlated instantaneously. Color scale represents both correlations, binned into quantiles (e. g. purple – high positive correlation of NDVI with both $T_{air}$ and Prec, green – high negative correlation of NDVI with both $T_{air}$ and Prec). Data points where NDVI<0.2 were excluded to avoid influence of inactive vegetation or non–vegetated time points. **b+c.** Correlations for different land cover classes (GLC2000) in the seasonal (b) and longer–term (c) scale.

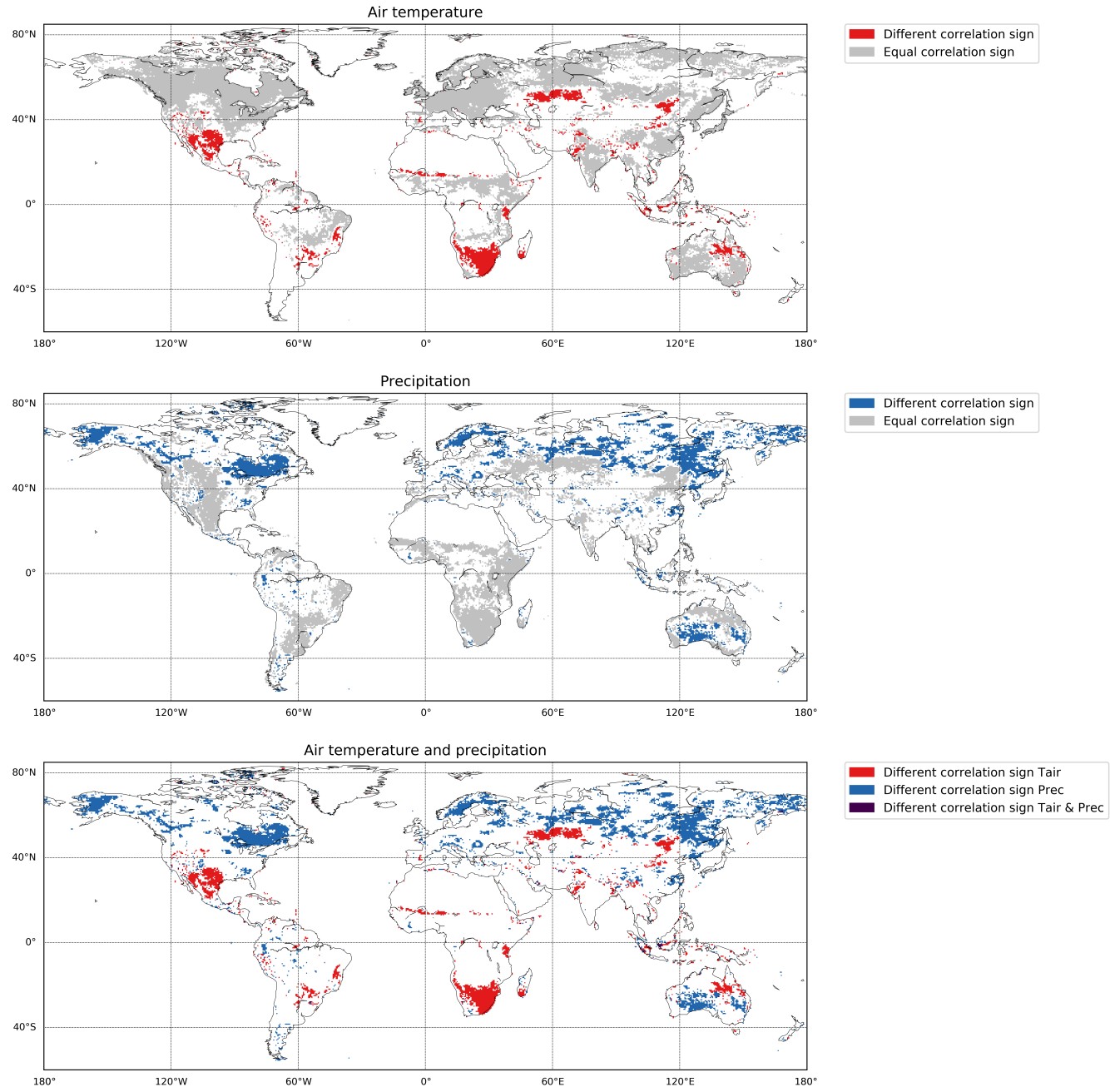

**Figure 4. Global comparison of differences in the sign of correlation between annual and long-term scale for NDVI and air temperature (top), NDVI and precipitation (center), and summary of both (bottom).** Areas in which the sign of correlation is inverted between seasonal and longer–term scales are highlighted in color, areas where the sign of correlation is identical between scales are highlighted in grey (a,b). Areas with correlations between –0.2 and 0.2 were not considered.

## 4 Discussion

In this study, we present a global characterization of biosphere variability at multiple time scales from weeks to decades where a natural surface classification emerges. We find that a substantial fraction of terrestrial ecosystems is characterized by either short– or longer–term NDVI oscillations (27% of variance globally). The grid cells dominated by longer–term oscillations
in NDVI concentrate mainly in semi–arid shrublands, and the short–term dominated grid cells in equatorial latitude forests. Patterns in NDVI, air temperature and precipitation variability are spatially heterogeneous: The classification of co–dominant oscillations is particularly homogeneous for temperate and boreal regions, while the tropics exhibit complex patterns of co–dominating time scales in vegetation and climate. This lack of correspondence in dominant temporal oscillations suggests that certain modes of variability in ecosystem–atmosphere interactions can be potentially induced by different exogenous, or
even endogenous, dynamics. This picture is further differentiated by the finding that correlations between NDVI and climate variables differ between time scales. This highlights the need to assess vegetation sensitivity to climate specifically on different scales in order to understand complex patterns of atmosphere–biosphere interactions in time, where also confounding factors should be considered.

### 4.1 Comparison across time scales points to complex temporal signatures

The combination of time scale–specific classification (Fig. 2) and correlation (Fig. 3) allowed us to characterize the major scales of vegetation variability in relation to climate. The classification provides an additional layer of ecosystem characterization beyond common classifications such as land cover classes or the effective Köppen–Geiger climate classifications (Kottek et al., 2006; Koeppen, 1900; Geiger, 1954) which only consider seasonality besides mean climate states, increasing our understanding of dynamic vegetation properties across time scales. The complementarity of this data–driven classification of
vegetation dynamics, extracted from the time series and summarized in the co–oscillation classification, is supported by the low spatial association calculated from the V–measure. Our findings show that the dominant oscillation of NDVI is often, but not always, related to dominant oscillations of $T_{air}$ and Prec (Fig. 2). For example, most of the land surface is dominated by annual oscillations in NDVI and $T_{air}$, combined with either seasonal or short–term dominance of Prec (AAA and AAS classes). In many of these regions, air temperature alone, or both air temperature and precipitation are limiting factors for
plant growth (Nemani, 2003; Seddon et al., 2016) and thus expected to drive vegetation dynamics. In contrast, heterogeneous spatial patterns are observed in equatorial and semi–arid regions, where different dominant scales of oscillation are found for NDVI and climatic variables. Here, the relationship between variables may depend on additional factors, and/or scales may show interactive effects. In the tropics, radiation is proposed to be one of the main drivers of NDVI (Nemani, 2003; Seddon et al., 2016), which could partially explain the lack of temporal coherence between NDVI, $T_{air}$ and Prec. Dominant short-term
oscillations of NDVI (SSS, SAS, SAA) might be explained by climate intra–seasonality in the tropics due to the Madden Julian Oscillation (MJO). The MJO is defined as anomalies in the atmospheric pressure between 10°N and 10°S in the Indian Ocean region that propagates eastward to the Eastern Pacific Madden and Julian (1971). Depending on the region and phase, its oscillatory period ranges between 20 and 90 days. MJO is considered the dominant component of intra–seasonal climate vari-

ability in the tropics (Zhang, 2013). We see MJO as one feasible driver of short–term NDVI opscillations through alterations of precipitation and temperature (Zhang, 2013; Hidayat, 2016; Mayta et al., 2019). However, MJO impacts, teleconnections and predictability are still insufficiently understood (Zhang, 2013; Wang et al., 2019). Short-term oscillations of vegetation in those regions need to be further investigated; including other sources of intra–seasonal variation, connections with climatic events, and data constraints. Additionally, regional analysis at higher spatial resolution might reveal details in local climatic variability, and other non–climatic processes such as land use change or crop rotations, among others. Comparing variables across multiple time scales can point to areas with complex temporal signatures that require further attention.

### 4.2 Non–dominant sub–signals reveal short– and longer–term ecosystem dynamics

From assessing relationships among variables on multiple time scales, we conclude that (i) non–dominant sub–signals in climate variables may dominate the biospheric response, (ii) possible links may exist between short–term and longer–term scales, and (iii) correlations of NDVI with climate variables differ between scales.

The dominance of long–term NDVI in semi–arid regions coincides with strong correlations of longer–term NDVI with Prec (positive) and $T_{air}$ (negative). This indicates that longer–term variation in precipitation exerts strong influence on NDVI variability in these regions despite contributing a minor portion of precipitation variation itself. Overall, longer–term correlation of precipitation with NDVI is higher than seasonal correlation (by at least 0.2) in 73% of the area classified as LAS where simultaneously longer–term variance of precipitation itself contributes $<20\%$ variance (3'939'362 km$^2$). Due to their highly plastic inter–annual vegetation dynamics, semi–arid ecosystems exert strong influence on inter–annual variability of the land CO$_2$ sink (Ahlstrom et al., 2015; Poulter et al., 2014; Zhang et al., 2016). Longer–term correlations between variables also show broad patterns related to temperature–induced greening in the Northern latitudes (Pan et al., 2018; Keenan and Riley, 2018; Zhu et al., 2016; Park et al., 2016). This is in agreement with previous findings using higher resolution data (Clinton et al., 2014). Thus, non–dominant sub–signals in climate variables may dominate the biospheric response, stressing their possible long–term impact on vegetation dynamics.

Vegetation may respond to inter–annual climate variation on both intra– and inter–annual scales (Meir et al., 2018). Such inter–annual climate variation may occur e. g. in the form of precipitation variation or periodic atmospheric fluctuations like El Niño Southern Oscillation (ENSO, Poveda and Salazar, 2004; Kogan and Guo, 2017; Liu et al., 2017), the Pacific Decadal Oscillation (Chen et al., 2017), or Indian Ocean Dynamics (Hawinkel et al., 2015). As a prominent example in our study, for semi–arid regions both short and longer–term correlations indicate a strong coupling to variations in water availability for shrublands and herbaceous land cover. These results harmonize with observed fast response of vegetation to water deficit in arid and semi–arid regions (Vicente-Serrano et al., 2013; Wang et al., 2016), as well as the observation of strong water memory effects in these regions (Liu et al., 2018). Some of these patterns match regions where vegetation is stressed during ENSO events due to precipitation decrease (Ahlstrom et al., 2015; Kogan and Guo, 2017), generating a possible link between short–term and longer–term scales. Previous studies suggest that climate forcing on one time scale can be amplified or dampened in corresponding vegetation responses (Stoy et al., 2009), or transferred to another time scale (Katul et al., 2001), preserving the

system's entropy but creating complex interactions across scales. This highlights the need to further investigate interactions between different time scales globally in long–term EO records.

Finally, for some regions the correlation of variables can differ between time scales. In southern Africa, for example, this may be due to a pronounced temperature–dependent annual cycle of vegetation, but longer–term negative effect of warming temperatures on vegetation productivity. Thus, time series decomposition offers important differentiation of atmosphere–biosphere co–variation across scales. This may serve as a platform for generating hypotheses in areas where contrasting dominant oscillations and/or correlations across scales are observed.

## 4.3    Differences between land cover classes highlight the tropics

By characterizing the temporal behaviour of NDVI and climate, we observed different vegetation dynamics between land cover types. Differences in power spectra between plant functional types have been shown before on shorter time scales with flux data (Stoy et al., 2009). Assessing this phenomenon globally, we find both homogeneous and heterogeneous behaviour within land cover types, showing non–trivial global patterns of the influence of land cover and climate on vegetation variability across scales. For example, 36% of Evergreen broadleaf forests are dominated by short–term oscillations in NDVI, while other forest types are dominated almost exclusively by seasonal NDVI oscillations. Indeed, the most heterogeneous patterns of co–dominating oscillations and correlations were found for tropical regions, within and across continents. In African tropics, NDVI is predominantly seasonal and correlation of NDVI with precipitation is always positive, while in most of the remaining tropics, NDVI is dominated by short–term oscillations and shows predominantly negative correlation with Prec on seasonal scale in the central Amazon and Southeast Asian tropical forests. This could be explained by different amounts of mean annual precipitation (MAP) falling in these regions, which cause a pronounced wet–dry seasonality in Africa and the central Amazon, but not in northwest or outer regions of the Amazon and SE Asia where MAP is in excess of annual vegetation water demand (Guan et al., 2015). In such areas, correlation with Prec may e. g. become negative when water is already in excess and clouded/rainy seasons cause limitation in radiation available for plant growth. Similarly, temperature is not usually limiting canopy development in the tropics (rather the contrary, Huang et al. (2019)), which may explain negative correlations with $T_{air}$. As NDVI saturates over regions of dense vegetation, results in the tropics need to be interpreted with caution, and negative correlation with $T_{air}$ could alternatively be explained by underestimation of the seasonal cycle over tropical EBF. In fact, negative correlations with $T_{air}$ were observed less frequently when repeating the analysis with MODIS EVI (Sup. Fig. S11), indicating that saturation of NDVI against plant productivity might affect our results in densely vegetated areas such as the tropics. Overall, despite known drawbacks of NDVI as proxy for plant productivity, the long–term NDVI record generally agrees well with results obtained from the considerably shorter EVI time series, suggesting that it is a good proxy for vegetation activity across time scales over large parts of the global land surface.

It is relevant to emphasize that our results might be affected by noise related to continuous presence of clouds in the tropics and other atmospheric artifacts. To estimate this effect, we excluded pixels with low number of direct observations and recalculated the dominant oscillation regime. Short–term oscillations were most affected in this analysis, but roughly 80% were

410 consistently dominated by short–term oscillation under the strictest scenario, providing higher confidence in our results for the tropics (Sup. Fig. S5).

The observed scale–specific patterns highlight the need to assess dynamic vegetation properties in time as differentiating factors beyond land cover type and mean climate.

## 4.4 Results prove robust against changes in data source and decomposition method

We used long–term GIMMS NDVI records in combination with Fourier transformation in this analysis, well aware of their potential limitations (van Leeuwen et al., 2006; Beck et al., 2011; Fensholt and Proud, 2012; Pinzon and Tucker, 2014). Further consolidating our results against methodological artifacts of data source and decomposition method, we found that results were not broadly affected or conclusions changed when repeating analyses with MODIS NDVI and EVI or Empirical Mode Decomposition (EMD), when excluding gap–filled values as discussed above, or when testing the effect of land use change on 420 decomposed oscillations. Specifically, the analysis of MODIS NDVI and EVI returned similar classification of dominant time scales in vegetation (Sup. Fig. S6). Although short–term oscillations in tropical NDVI may partly reflect noise introduced by cloud cover, heavy aerosol conditions and biomass burning, our results based on EVI, which is less sensitive to aerosols and haze (Miura et al., 2012), resulted in even more, rather than less, pixels being classified into the short–term oscillation regime (Sup. Fig. S6). However, dense clouds are still a limitation when optical remote sensing data is used. EMD decomposition 425 consistently reproduced results of FFT in space and time for all variables (Sup. Fig. S15–18). Excluding gap–filled values originating from snow or cloud inference from the GIMMS NDVI dataset changed the dominant oscillation for only up to 2.3% of pixels overall, and 20% of short–term classified pixels, when the strictest threshold was applied (Sup. Fig. S5). Land cover and land use change were hardly detectable at the coarse spatial resolution of 0.5° employed, and had a minor effect on the distribution of signal variance to the different time scale (Sup. Fig. S7). In summary, our results proved robust against data 430 source and decomposition method.

## 4.5 Limitations and Outlook

The current study presents a first global characterization of atmosphere–biosphere variability at multiple time scales from weeks to decades. We chose the longest available satellite–retrieved time series of vegetation, GIMMS NDVI, to be able to assess relations of atmosphere–biosphere co–variability over more than three decades. We find heterogeneous temporal patterns 435 of biosphere–climate responses across time scales. Known limitations of NDVI include saturation effect at high canopy cover, especially relevant in the tropics, as well as influence by soil reflectance in sparsely vegetated areas. These effects could thus influence our results and the emerging patterns should be compared with newer satellite products such as sun–induced fluorescence (SIF), which are coupled more directly to plant physiology and photosynthesis (Badgley et al., 2017; Koren et al., 2018), but are only available for short time periods. Considering further variables influencing vegetation dynamics, such as 440 radiation, cloud cover, soil moisture, fires, or storms could bring additional insight into the drivers of vegetation dynamics, especially for poorly explained regions in the current analysis, such as the tropics. In future studies, longer–term climate

signals could be compared with climate oscillations such as ENSO to gain further understanding of their effect on long–term ecosystem variability.

Analysis of time lag effects between atmospheric forcing and vegetation response may bring additional valuable insights into ecosystem functioning, yet assessing meaningful time lags across time scales is challenging due to a variety processes involved. Plausible time lags from months to years have been suggested between climate forcing and vegetation response and/or ecosystem carbon exchange through direct and indirect effects (e. g. Braswell et al., 1997, 2005; Vukićević et al., 2001; Krich et al., 2019; Kraft et al., 2019; Papagiannopoulou et al., 2017). Assessing lagged vegetation responses across time scales may help to disentangle such co–existing time lags to form a global, time–scale–resolved picture of vegetation responses to climate. To account for the confounding effect of autocorrelation and spurious links between variables, methods like causal inference (Runge et al., 2013, 2019; Krich et al., 2019) should be applied in order to retrieve causal time lags between variables.

Our analyses are conducted at 0.5° spatial and 15–daily temporal resolution, which may obscure short–term and local vegetation–climate relations, and instead only provide average relationships of variables within each grid cell. Our analyses may thus not be representative in heterogeneous landscapes such as coastlines or mountains. Regions standing out through heterogeneous patterns, such as the Amazon, should be further investigated regionally at higher temporal and spatial resolution whenever consistent data streams permit this, to better understand local influence of climate, vegetation and topography on atmosphere–biosphere co–variation. Recently, studies in the Amazon based on products such as SIF have detected differences in vegetation anomalies within the basin during El Niño events (Koren et al., 2018). The identified asymmetry in the East–West gradient coincides with observed changes in temperature, soil moisture and GRACE–derived water storage. Our results pave a way for better understanding the spatial heterogeneity of ecosystem responses to climate variability (van Schaik et al., 2018). Here, assessing temporal patterns beyond correlation (c. f. Wu et al., 2015) will provide additional insights into the temporal evolution of vegetation dynamics, and the carbon cycle variability.

## 5 Conclusions

In conclusion, decomposing vegetation and climate time–series into discrete subsignals allows us to disentangle atmosphere–biosphere oscillations from short– to longer–term time scales. A key finding is that short–term and longer–term modes of variability can dominate regional patterns of ecosystem dynamics: 18% of land area is effectively characterized by intra–annual variability, and 9% by longer–term modes of NDVI. We derived a global map of dominant patterns of vegetation–climate co–variability on multiple time scales. The emerging classification of variability regimes allows us to generate new hypotheses on land-atmosphere interactions. In particular, we can now delineate areas with complex spatio–temporal vegetation signatures. For example, tropical evergreen forests are dominated by short-term oscillations (36%), while shrublands and herbaceous land cover make up >90% of area dominated by longer-term NDVI, suggesting important roles in intra– and inter–annual biosphere dynamics of these land cover classes. Importantly, changing correlations of NDVI with climate across time scales suggest that climate sensitivities of vegetation can vary with time scale. Globally, 15.4% of land area show opposing correlation of NDVI to $T_{air}$ between annual and long–term modes of variability, while 27.3% show opposite correlations of NDVI and Prec. These findings underline the relevance of advancing our understanding of scale-specific climate sensitivities. In Southern Africa, for instance, the relation of vegetation to temperature is inverted across scales, as well as in parts of Australia where the same is true for precipitation. Differentiating such responses is essential to fully comprehend long–term biosphere dynamics and project them into the future. Understanding the interaction of climate and vegetation on separate time scales, warranting that independent processes are not obscured by dominant ones, is essential in times where extreme climate conditions of increasing frequency exert a repeated perturbing influence on ecosystem dynamics (Defriez and Reuman, 2017). This needs to be considered for short– and long–term vegetation modeling under changing climate scenarios.

*Code and data availability.* All remote sensing data is publicly available from the respective supplier. The co–oscillation classification (Fig. 2) is available as NetCDF file from doi:10.5281/zenodo.3611262. The code underlying all primary figures is made available as supplementary notebook from doi:10.5281/zenodo.3611262 and in the Supplementary Material.

485 *Author contributions.* L.E. and N.L. have contributed equally to the paper and performed all analyses. The study was designed by N.L., M.D.M, L.E. with input from all other authors. L.E. and N.L. wrote the manuscript with substantial input from all other authors. F.C. and F.G. implemented the EmpiricalModeDecomposition.jl and libeemd.jl packages in julia.

*Competing interests.* The authors declare that they have no conflict of interest.

*Acknowledgements.* This paper has been realized within the "Earth System Data Lab" project funded by the European Space Agency. L.E.
acknowledges the DAAD for kind support. N. L. acknowledges the TUM Graduate School for kind support. L.E. and N.L. acknowledge the International Max Planck Research School for Global Biogeochemical Cycles for continuous support. The authors acknowledge Lina Fürst for initiation of the pre–study laying the foundation for this project. The authors acknowledge support from Ulrich Weber for data management and preprocessing, and the Biogeochemical Integration Department at the Max Planck Institute for Biogeochemistry for continuous support. This project was partially funded through DFG (German Research Foundation) project HyperSense (grant No. TH 1435/4-1).

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
