# Peer review of "Towards a global understanding of vegetation–climate dynamics at multiple time scales"

_Biogeosciences, 2019_

## Referee Comment (RC1) · Anonymous Referee #1 · 22 Oct 2019

This paper analyses temporal variability in vegetation greenness (NDVI), air temperature (T), and precipitation (P) over broad range of time scales and over the entire terrestrial landmass at 0.5 degree resolution. The overall purpose of the paper is to identify vegetation-climate co-variations. I am impressed by the scope of the work presented in this paper and by the very detailed description of the data and methods.

The use of the NDVI as a metric of vegetation greenness could be controversial but the authors gave a good argument in favor of their choice. The choice of 0.5 degree as the spatial resolution has not been explained, I suspect that it was needed for averaging purposes due to lapses in data coverage at the higher resolution. However, it is important to remember that such resolution amounts to about 55 km on the equator. This is a significant area over which to average NDVI, T, and P, especially in the mountainous and coastal areas where significant spatial variability of these quantities are expected at that spatial scale. A decision to work on the lat-lon grid is also left without comments. Using the lat-lon grid for global studies is questionable even if it is common in the literature. This problem is multiplied by the use of large grid cells, the shape and areas of cells at different latitudes change significantly. Projection to other grids, more appropriate to global analysis, is possible. Authors should discuss their choice of using the lat-lon grid with large grid cells. The Fourier transform method of dividing signal into short-term, seasonal, and long-term is standard and effective.

The global maps of NDVI, T, and P variance decomposed into three regimes of temporal variability (short, seasonal, and long) shows (Fig.1) mostly what I would expect with the exception of short-term variability of NDVI. I think it would be useful if the paper would mention that seasonal variability of NDVI and T is a "default" variability – something expected. Long term variability of these two variables is something that we can understand, but short-term variability is surprising, especially for the NDVI which should have more "inertia" than T. On the other hand, short-term variability of P is expected as we all know from experience, seasonal variability of P is also expected and long-term variability could be understood. More generally, the paper describes results in detail but lacks commentary in a spirit of what I outlined above. This is not that we don't have any expectations of how the results should look like, so it would be effective to stress the unexpected parts of the results.

Authors have designed a clever scheme to show "co-oscillation regimes." It classifies grid cells' into 11 classes based on co-occurrence of dominant temporal variability regime (S, A, or L) for each variable (NDVI, T, P). This yields a compact and easy to understand map (Fig.2). Again, I argue that it would be useful to stress that AAS and AAA classes are expected by default – something that we would predict on the basis of prior knowledge without analysis presented in this paper. Thus focus should be on the remaining classes. The LAS class can be easily explained, these are semi-desert or shrubland areas with only slightly seasonal-dependent vegetation and sporadic rainfall.
On the other hand SSS, SAA, and SAS classes are difficult to understand. Authors do not try to understand them; instead they just say that this points to a complex climate-vegetation dynamics. I think that this is a way of saying "we tried to understand them but failed" or "we did not try to understand them in this paper". Both statements are acceptable but should be stated clearly. To me these classes are difficult to accept, for example, what does that means that some regions in Indonesia are characterized by predominantly short-term variability of vegetation and temperature? It seems counter-intuitive and contrary to prior knowledge. Could it be a data artifact, or method artifact? One possible explanation is that the short-term temporal variability include all periods < 0.9 year, so maybe in these regions the actual variability is ∼0.9 year which would be more intuitive.

Authors compare their map of "co-oscillation regimes" with the map of land cover (GLC) and, separately, with the map of climate (Koppen-Geiger or KG). The reason for such comparison is not clearly explained. On one hand authors describe their classification as new and different (from climate and land cover classifications), but, on the other hand, they look for similarities with those maps. I am not saying that such comparison is uninteresting, just that the rationale is not explicitly given. Also, maps could be compared using methods especially designed for such purpose (see International J. of GIS, 2018, v32(12), pp. 2386-2401).

Authors also constructed maps (Fig.3) on which a color (organized into a bivariate table) indicates correlation between NDVI and T signals, and, at the same time, a correlation between NDVI and P. As in the previous two figures, I found that the text should stress that seasonal cycle is a default and short and long term cycles are the "new" results. In this case, however, because of the uniqueness of the bivariate map, even the seasonal map is new and could use more explanation going beyond just describing what we already see in the map.

Overall, I find this paper to be solid, very interesting and illuminating; however, it is, to my taste, overly skewed toward technical description at the cost of not providing

plausible explanations to some of the more unexpected results.

---

## Short Comment (SC1) · 9 Nov 2019

**Disclaimer**: This review was written by MSc student Maaike de Boer  as part of her course work on "scientific reviewing", under supervision of dr Arnold Moene from Wageningen University. The comments were submitted because they can contribute to the scientific process, and because they contain helpful questions and suggestions for the authors. Although the structure of this review follows the formal conventions, it is thus not a solicited peer-review from the editor of ACPD.

In the paper *Towards a global understanding of vegetation-climate dynamics at multiple time scales*, Linscheid et al. describe a novel approach to study the dynamics between vegetation and climate at multiple time scales. Due to long-term Earth observations (EOs), ecosystem analyses are possible at time scales of over thirty years. In this study, global NDVI is correlated with two climate variables, air temperature ($T_{air}$) and precipitation (Prec), and studied at short-term, seasonal, and long-term time scales. The different time scales were determined using Fast Fourier Transformation (FFT), and linear correlation methods were used for the correlation. Bivariate colour maps were used to present the correlations. The effect of land use change was also taken into account, to see its effects. The authors did a comparison between the techniques Fourier Transform and Empirical Mode Decomposition, to see whether the latter more data-adaptive technique leads to different insights.

The paper was written clearly and is well-structured, and the figures were clear and supportive of the paper. Using bivariate colour maps to present the correlations is an excellent choice for the variables considered in this research (Teuling et al., 2011). Similar research has been done before, but only for specific regions (Martinéz and Gilabert, 2009; Canisius et al., 2007; Hawinkel et al., 2015) or not taking co-interpretation with climate variables into account (Pan et al., 2018), which is why the findings in this paper are novel and valuable. The paper fits the scope of the journal *Biogeosciences* very well, but some revision needs to be done before the paper is ready for publication. The paper overall is good, but revisions could be made on several topics.

I recommend the authors to revise some parts of the paper. Specifically, in sections 2.1, 2.2, 2.3, 2.4 and 2.5 I have some major revisions. I also have some further minor and very minor revisions. I will go into further details on those revisions below.

Major comments:

− In section 2.5 the method for the correlations between variables at each time scale is discussed. Correlation of the time-scale specific sub-signals of NDVI, $T_{air}$ and Prec was done using Pearson's correlation coefficient, Spearman correlation, and partial correlation. All three of those methods assume linear correlation. However, the authors do not specify anywhere in the paper why, or even if, linear correlation can be assumed for the data they are analysing. This assumption should be elaborated on, because if linear correlation could not be assumed, these correlation analyses would not be a possible method. The choice of linear correlation affects results section 3.4 and Figures 3 and 4, thus revision is needed. My proposed revision entails the authors explaining which assumptions are made to allow for linear correlation. It is also advisable to discuss the consequence of assuming linear correlation in the discussion, to evaluate its effect on the results of the study.

− In section 2.1 the data that is used for the study is elaborated on. A global gridded dataset of NDVI at 0.5° spatial resolution is used. $T_{air}$ and Prec data is obtained at 0.083° spatial resolution. To match the NDVI data, the data for $T_{air}$ and Prec were aggregated to 0.5° spatial resolution. For $T_{air}$ this was done by averaging, and for Prec this was done by summation. However, detail on extreme values for both $T_{air}$ and Prec is lost by this aggregation. For $T_{air}$, local extreme values will be lost due to averaging over a larger area, while for Prec local extreme values will be smoothened out due to summation over a larger area. Upscaling of the $T_{air}$ and Prec data is part of the pre-processing of

the data, thus it has effects on all results and conclusions. A revision is recommended to include a quantification of the sensitivity of the resulting classification in regimes to the spatial averaging. Furthermore, NDVI data is available at 0.05° spatial resolution (National Center for Atmospheric Research Staff), so the upscaling of the $T_{air}$ and Prec data to 0.5° is not necessary. There could be many reasons for not choosing the higher resolution 0.05° data, for instance due to the 0.5° data being more manageable. However, there is no discussion on these possible reasons. Revision is therefore advised to include a discussion about the choice for 0.5° rather than 0.05° spatial resolution data.

− In section 2.2 the time periods that are used to reconstruct the different time scales are described, stating that similar frequency bins have been used in previous studies. The literature referenced however, by Mahecha et al., has no mention of these similar frequency bins. The article only states that the annual-seasonal was defined as periods in the interval 0.375-1.25 years. I could not obtain the other literature that was referenced for these similar frequency bins, a thesis by Fürst. Basing the time periods only on the annual-seasonal interval mentioned in the article by Mahecha et al. without further explanation is not valid. This section should be revised, and argumentation for the chosen time periods should be included, as well as a quantification on how the conclusions are influenced by these time period boundaries.

− The time series decomposition is done by Fast Fourier Transformation. However, Mahecha et al. (2010) state that classical Fourier decomposition is not the best method to segregate data into different temporal scales. Discrete Wavelet Transforms (DWT) or Empirical Mode Decomposition (EMD) are mentioned as better alternatives, because those methods do not assume a fixed superposition of weighted sines and cosines. The authors do a comparison of Fourier Transform with EMD in section 2.7 of the methods, and they discuss the difference in results between the two techniques in section 4.4, but it is recommended to include an explanation on the choice for FFT as method for the main research.

− In section 2.5, lines 124-125, it is stated that NDVI was lagged one time step (15 days) behind Prec in order to allow response time of vegetation to changes in water availability. However, Jamali et al. (2011) found a response time of 8-40 days of vegetation indices to rainfall changes for six sites in Africa. This response time is much longer than the 15 days assumed in this paper. The study by Jamali et al. was conducted in Africa, so perhaps in more moderate climates, the response time would be closer to 15 days. A revision is recommended to include a motivation for the 15 days response time chosen for this study, to discuss on what literature this choice was based and whether this is a valid choice for a global estimate.

− In section 2.4 the method for determining the variance per time scale and co-oscillation regime is discussed. It is explained that 64 possible combinations of oscillation regimes are possible, of which only 26 occurred. For simplicity, the analysis focussed only on the 11 most abundant oscillation regimes, which comprise 99.7% of pixels. The only argumentation given by the authors for choosing these 11 most abundant oscillations is for simplicity, which is not a concrete reason to make this choice. The choice to take only the 11 most abundant oscillation regimes into account affects results sections 3.2 and 3.3, as well as Figures 2 and 3. I therefore recommend the following revisions: the authors should elaborate on what they mean by simplicity. They should also specifically discuss why the eleven most abundant oscillation regimes are chosen, rather than a more logical and round number such as 10 or 15. The same goes for the percentage of pixels, why choose 99.7% rather than for example 99.5% or 99%?

Minor comments:

− No information is provided on data processing of NDVI to minimise effects by external factors. Some external factors that could have an effect are mentioned by Zeng et al. (2013): solar zenith

angle or volcanic stratospheric aerosol effects from major volcanic eruptions. Please include a discussion on possible effects by external factors.

− p5, lines 126-128: only the seasonal and longer-term oscillations are compared, and no motivation is provided for not including short-term oscillations in this comparison. Please elaborate on why short-term oscillations are not included in the comparison.

− p9, Fig 2a: it is not specified what the white colours on the map correspond with. It can be assumed that these correspond with NA, but this should be mentioned in the caption.

− p12, Fig. 3a: just as for Fig. 2a, no specification on what the white colours on the map correspond with. The caption mentions that NDVI<0.2 were excluded, but it is not mentioned that these are shown in white on the map.

− p13, Fig. 4: same as for Fig. 2a and Fig. 3a. The caption says that areas with correlations between -0.2 and 0.2 were not considered, but it is not mentioned that these are shown in white on the map.

Very minor comments:

− p4, line 110: change notation from "75'871'486 km$^2$" to "75,871,486 km$^2$".
− p7, Fig. 1: in the caption, change "seasonal(annual)" to "seasonal (annual)".
− p9, Fig. 2: in the caption, "DBF" is mentioned, but in the figure it is called "DBF_open". Please change one of the two, so they correspond with each other.
− p10, line 219: change "At" to "at", because after a colon.
− p10, line 228: change "While" to "while", because after a colon.
− p10, line 237: change "For" to "for", because after a colon.
− p10, line 239: change "northern" to "Northern".
− p11, line 269: change "Tair" to "$T_{air}$".
− p11, line 270: add a comma between "method" and "a".
− p12, Fig. 3: remove the second "." in the bold part of the caption.
− p13, Fig. 4: change "Tair" to "$T_{air}$" in the legend of the bottom map.
− p14, line 274: change "characterised" to "characterized" for consistency, because in the rest of the paper American spelling is used.
− p14, line 285: change "point" to "points", because it is a singular verb.
− p14, line 293: add a comma between "precipitation" and "are".
− p15, line 308: change notation from "3'939'362 km$^2$" to "3,939,362 km$^2$".
− p18, line 400: change "Tair" to "$T_{air}$".

**References:**

• Teuling, A. J., Stöckli, R., and Seneviratne, S. I.: Bivariate Colour Maps for Visualizing Climate Data, International Journal of Climatology, 31, 1408–1412, https://doi.org/10.1002/joc.2153, 2011.

• Martínez, B. and Gilabert, M. A.: Vegetation Dynamics from NDVI Time Series Analysis Using the Wavelet Transform, Remote Sensing of Environment, 113, 1823–1842, https://doi.org/10.1016/j.rse.2009.04.016, 2009.

• Canisius, F., Turral, H., and Molden, D.: Fourier Analysis of Historical NOAA Time Series Data to Estimate Bimodal Agriculture, International Journal of Remote Sensing, 28, 5503–5522, https://doi.org/10.1080/01431160601086043, 2007.

- Hawinkel, P., Swinnen, E., Lhermitte, S., Verbist, B., Van Orshoven, J., and Muys, B.: A Time Series Processing Tool to Extract Climate-Driven Interannual Vegetation Dynamics Using Ensemble Empirical Mode Decomposition (EEMD), Remote Sensing of Environment, 169, 375–389, https://doi.org/10.1016/j.rse.2015.08.024, 2015.
- Pan, N., Feng, X., Fu, B., Wang, S., Ji, F., and Pan, S.: Increasing Global Vegetation Browning Hidden in Overall Vegetation Greening: Insights from Time-Varying Trends, Remote Sensing of Environment, 214, 59–72, https://doi.org/10.1016/j.rse.2018.05.018, 2018.
- National Center for Atmospheric Research Staff (Eds). Last modified 15 Mar 2018. "The Climate Data Guide: NDVI: Normalized-difference-vegetation-index: NOAA AVHRR." Retrieved from https://climatedataguide.ucar.edu/climate-data/ndvi-normalized-difference-vegetation-index-noaa-avhrr.
- Mahecha, M. D., Fürst, L. M., Gobron, N., and Lange, H.: Identifying Multiple Spatiotemporal Patterns: A Refined View on Terrestrial Photosynthetic Activity, Pattern Recognition Letters, 31, 2309–2317, https://doi.org/10.1016/j.patrec.2010.06.021, 2010a.
- Fürst, L. M. M.: Characterizing Global Spatiotemporal Patterns of the Fraction of Absorbed Photosynthetically Active Radiation, Diploma Thesis, University Bayreuth, Bayreuth, 2009.
- Jamali, S., Seaquist, J., Ardö, J., Eklundh, L. (2011). Investigating temporal relationships between rainfall, soil moisture and MODIS-derived NDVI and EVI for six sites in Africa, Conference Paper for the 34th International Symposium on Remote Sensing of Environment.
- Zeng,F.-W., Collatz,G., Pinzon,J., and Ivanoff,A.: Evaluating and Quantifying the Climate-Driven Interannual Variability in Global Inventory Modeling and Mapping Studies (GIMMS) Normalized Difference Vegetation Index (NDVI3g) at Global Scales, Remote Sensing, 5, 3918–3950, https://doi.org/10.3390/rs5083918, 2013.

---

## Short Comment (SC2) · 13 Nov 2019

Review of *Towards a global understanding of vegetation–climate dynamics at multiple time scales* by *Linscheid et al.* by reviewer Le Lu

Linscheid et al. provide a global assessment of vegetation-climate variability at multiple time scales. Motivated by the existing knowledge gap regarding how the global terrestrial biosphere responds to multi-timescale variability of climate, the study aims to explicitly examine the variability of biosphere and climate time series, explore spatial variability of vegetation-climate dynamics at each time scale, assess the potential link between the resolved temporal patterns and traditional land cover classification and compare correlations of climate and vegetation across multiple time scales. By decomposing over 30 years of remote sensing records of NDVI and climate time series with Fast Fourier Transformation (FFT), variances in each variable contributed by different time scales from short-term oscillations to seasonal and long-term trend were compared and co-oscillations between NDVI and climate variables (i.e. temperature and precipitation) were discussed. Moreover, the oscillations regimes were compared with traditional classifications of land covers and climate zones. For every time scale, the researchers also examined the correlations of NDVI with climate variables. Finally, the potential effects of land cover change, limitations of FFT and NDVI on time series decomposition were also considered.

The results show heterogenous response by vegetation to climate variables both temporally and spatially. Section 3.1 presents that NDVI and temperature variability is dominated by seasonal cycle whereas 52% of the variance in precipitation is contributed by short-term oscillations on a global scale. However, differences exist as the long-term oscillation in NDVI is strongly associated with shrub and herbaceous land cover types. In section 3.4, a comparison of correlations of NDVI with climate variables suggests that correlations strongly vary with time scales and space. In South Africa, correlation with temperature is negative on short-term scale and long-term scale but positive on seasonal scale. Different correlations patterns are observed among tropical forest regions. Section 4.4 concludes that the results are proven to be robust by repeating the analysis with Enhanced Vegetation Index (EVI) and NDVI from the Moderate Resolution Imaging Spectroradiometer (MODIS) as indicators of vegetation dynamics or Empirical Mode Decomposition as the time series decomposition technique.

The study is, as its authors claim, the first of its kind. It is the first attempt to examining the global vegetation response to climate across multiple time scales. What is particularly strong about this study is that its methodology takes into account various sources of possible confounding factors that may compromise the results. As mentioned above, time series decomposition is repeated with a more data-adaptive approach. Considering the noise introduced by cloud cover, which NDVI is subject to, vegetation indices from MODIS are included as part of the repeated analysis so the results can be proven robust. The effect of land use and land use change (LULUC) is evaluated as well given the large time span that the study incorporates. Last but not least, the presentations of results are clear and easy to follow,

particularly figure 2a in which the dominant oscillations of NDVI, temperature, and precipitation are indicated by a three-letter scheme.

Overall, the study is well-thought-out and well written. This is a study of biosphere-atmosphere interactions, so it falls well within the scope of the journal. It serves as an important contribution to our understanding of past and current climate-vegetation interactions on separate times scales, and its findings may even help us project the future interactions. In the meantime, however, there are a few issues that I believe require further attention and revisions before publication. These issues are discussed below in detail.

**Major arguments**

1) The methodology of this study relies on the use of long-term remote sensing records of NDVI. It is good to see some limitations of remote sensing NDVI are addressed by comparing the results with alternative data source and vegetation index such as EVI and NDVI from MODIS. Yet other known issues of NDVI are not discussed in the manuscript.

   It is well documented that NDVI can be biased in regions where biomass is dense (Pettorelli et al., 2005). NDVI saturates as plants grow and thus is insensitive to dense biomass (Huete et al., 2002). the saturation of NDVI may cause underestimated seasonal cycles in tropics, where vegetation is typically dense. This potential outcome could be an alternative explanation to the contradiction between the results of this study and Huang et al. (2019) regarding the negative correlations with temperature in the tropics. Furthermore, a recent study, which is cited for other parts of the study, attributes the low response of vegetation to precipitation in African tropics as partly a result of saturation (Hawinkel et al., 2015), contradicting the claim that the correlation between NDVI and precipitation is almost always positive. In another previous study on global NDVI time series, tropical evergreen forest is even excluded because of the low signal/noise ratio associated with saturation (de Jong, Verbesselt, Schaepman, & de Bruin, 2011).

   Besides, NDVI is also influenced by reflectance from soil when the Leaf Area Index (LAI) is under 3 (Pettorelli et al., 2005; Huete, 1988). Specifically, darker soil substrates would result in higher values of NDVI for a given amount of vegetation. Huete (1988) notes that the NDVI suffers from soil reflectance issues across different soil types especially in global data sets. This is critical to this study and likely to have implications on its results since the global scale is exactly what it deals with and involves naturally various soil types. For example, the results of the study show strong associations between the long-term NDVI regimes and arid regions where vegetation is sparse, but this could be partly the result of varying brightness of soil controlled by soil moisture. Meanwhile, the short and long-term correlations with precipitation presented by the study may be overestimated due to the higher NDVI values associated with darker soil substrates led by precipitation. These unaccounted impacts on the current results, without further looking at the pattern of variations in soil brightness, is unpredictable. Nonetheless, it would either overestimate or underestimate the contribution of oscillations at certain time scales to the overall variability considering the influence of noise introduced by soil background on NDVI.

   To address the above limitations of NDVI, I recommend validating the results by repeating the analysis with another two vegetation indices just as what was done with MODIS EVI, at least for regions where vegetation is dense or sparse. The two indices are the Soil-Adjusted Vegetation Index developed by Huete (1988) and the Wide Dynamic Range

Vegetation Index by Gitelson (2004), which are designed to overcome the noise from soil background and saturation of NDVI, respectively. Both indices are modified versions of NDVI and require only the reflectance inputs in the same waveband as what NDVI does, so there is no need to introduce any new data. Although the values of different indices are not directly comparable, which may compromise the global scale that the study aims at, in this way it nevertheless would be clear whether the two limitations of NDVI have any major implications on the results, providing a more accurate characterization of vegetation-climate dynamics.

2) The study attempts to assess the influence of land use and land use change (LULUC) on its reported NDVI classifications and concludes no widespread effect in section 3.3. The cutoff of land use change is set at pixels with 25% change in fraction of vegetations. There is no reason given regarding why 25% is chosen, which makes me curious to learn what the results would be if a threshold lower than 25% is used, for example, 20% or 15%. More pixels would be inspected with a lower threshold, and maybe more pixels would reflect change in NDVI due to LULUC. In addition, the study does admit the absence of marked signs of LULUC is perhaps because of the coarse spatial resolution of the data. Despite knowing the limitation, the study still concludes there is LULUC has no widespread effect on the results, which I find very puzzling. To me, given the limitation in spatial resolution the best that could be concluded would be "no definitive conclusion can be given" instead of lines 224 and 225.

As a side note, some questions regarding the validity of the study may be asked because the coarse resolution of the data is proposed as the explanation for the absence of signs of LULUC in NDVI. If this explanation holds, would it be suggesting that the spatial heterogeneity of vegetation dynamics is underestimated by the coarse resolution?

Regarding the assessment of the effect of LULUC, I recommend to either give reasons for the chosen 25% cutoff or compare the results with a lower threshold value. For the validity of the conclusion on LULUC effect and even the entire study, it is perhaps wise to recognize the coarse resolution as a limitation to the study design and be clear about possible implications. This addition could be put into section 4.5 Limitations and Outlook to give a more balanced reflection on the methods and results.

3) Section 3.5 compares results from CEEDMAN with the FFT in order to prove the results are robust. However, hardly any justifications are given for carrying out this analysis other than lines $265 - 266$ "the data-adaptive empirical mode decomposition (EMD) could be expected to be better suited for exploring instationary ecological processes over time". It inevitably leaves readers to wonder about the relevance of this repetition, especially given that the results are largely the same. Moreover, the manuscript does not provide enough information about how the repetition was implemented. It recognizes the constraints posed by a higher spatial heterogeneity in implementing a global analysis with CEEDMAN, and it appears to allude the implementation includes a test case over Europe. However, it is not clear if any other regions were included as a part of the repeated implementation and if the results were based on any regions in addition to Europe.

I recommend the authors to be explicit in this section. Multiple papers are already cited regarding CEEDMAN, and I do not see why not convincing readers about the use of it by enumerate the benefits/advantages of this alternative method and the expected differences that the method may bring. It could also help readers better understand the actual yielded

differences shown in lines 268 to 270. It may also be useful to include more details as what the spatial scope of this repeated analysis instead of simply presenting the results of a test case over Europe.

**Minor issues**

1. In line 39 the authors state that only reflecting greenness is a limitation of NDVI, followed by multiple citations. After skimming the cited manuscripts, I found no relevant claims. One of the studies says the ability of NDVI to quantify greenness is what allows it to correlate with vegetation biomass and dynamics. In a rather thorough literature review I read regarding the use of NDVI for assessing vegetation response (Pettorelli et al., 2005), no relevant information is given about whether reflecting only greenness as a limitation of NDVI. It could be a factual error or the experience of the authors, but unfortunately no elaborations on this point are given. Please offer some evidence to support this claim.
2. In line 297, the authors suggest radiation as the main driver of NDVI in tropical regions. However, the interpretation of the source cited after the claim does not seem entirely accurate. Nemani (2003) says the tropical area "may have either a sustained dry season or nearly perpetual cloud cover that limits solar radiation", implying water availability and radiation may both be limiting factors in the tropics. However, the study only mentions radiation but not water availability to support an explanation for its results, which constitutes an inaccurate interpretation of the reference. I recommend rephrasing the sentence to accurately reflect the reference.
3. Page 9, figure 2a: No legends are given for land cover classes represented by the two types of red colors that are used in the figure.
4. Page 10, line 235: In other parts of the manuscript, it is being referred to as evergreen broadleaf forest (EBF), but in line 235 it is broadleaf evergreen forest. Replacing it with the acronym would do as this is not its first appearance.
5. Page 10, line 244: This is essentially a repetition of what was discussed in line 235 without adding any new information.

Other than the above issues, the work is well done. I am confident that it will be accepted after these issues are properly addressed.

**References**

de Jong, R., Verbesselt, J., Schaepman, M. E., & de Bruin, S. (2011). Detection of breakpoints in global NDVI time series. *De Jong, Rogier; Verbesselt, Jan; Schaepman, Michael E; de Bruin, Sytze (2011). Detection of Breakpoints in Global NDVI Time Series. In: 34th International Symposium on Remote Sensing of Environment (ISRSE), Sydney (AUS), 10 April 2011 - 15 April 2011, Online.*, online. https://doi.org/info:doi/10.5167/uzh-77356

Gitelson, A. A. (2004). Wide Dynamic Range Vegetation Index for Remote Quantification of Biophysical Characteristics of Vegetation. *Journal of Plant Physiology*, *161*(2), 165–173. https://doi.org/10.1078/0176-1617-01176

Hawinkel, P., Swinnen, E., Lhermitte, S., Verbist, B., Van Orshoven, J., & Muys, B. (2015). A time series processing tool to extract climate-driven interannual vegetation dynamics using Ensemble Empirical Mode Decomposition (EEMD). *Remote Sensing of Environment*, *169*, 375–389. https://doi.org/10.1016/j.rse.2015.08.024

Huang, M., Piao, S., Ciais, P., Peñuelas, J., Wang, X., Keenan, T. F., … Janssens, I. A. (2019). Air temperature optima of vegetation productivity across global biomes. *Nature Ecology & Evolution*, *3*(5), 772–779. https://doi.org/10.1038/s41559-019-0838-x

Huete, A., Didan, K., Miura, T., Rodriguez, E. P., Gao, X., & Ferreira, L. G. (2002). Overview of the radiometric and biophysical performance of the MODIS vegetation indices. *Remote Sensing of Environment*, *83*(1), 195–213. https://doi.org/10.1016/S0034-4257(02)00096-2

Huete, A. R. (1988). A soil-adjusted vegetation index (SAVI). *Remote Sensing of Environment*, *25*(3), 295–309. https://doi.org/10.1016/0034-4257(88)90106-X

Nemani, R. R. (2003). Climate-Driven Increases in Global Terrestrial Net Primary Production from 1982 to 1999. *Science*, *300*(5625), 1560–1563. https://doi.org/10.1126/science.1082750

Pettorelli, N., Vik, J. O., Mysterud, A., Gaillard, J.-M., Tucker, C. J., & Stenseth, N. Chr. (2005). Using the satellite-derived NDVI to assess ecological responses to environmental change. *Trends in Ecology & Evolution*, *20*(9), 503–510. https://doi.org/10.1016/j.tree.2005.05.011

---

## Referee Comment (RC2) · Anonymous Referee #2 · 17 Nov 2019

This is a nice analysis, although the significance for ecosystems and climate is not stated clearly anywhere in simple language. In contrast to other comments, this is not the first such analysis (see for example Braswell et al 1997 and Martinez and Gilabert 2009) as well as substantial other work on time scales using related data such as CO2 (Vukicevic et al 1997). This analysis, while acknowledging earlier work, doesn't build very much on insights from earlier work though it uses longer and more robust time series. Specifically, Katul's analysis (2001, I think) that showed that ecosystems may rectify short-term forcing onto longer time scales suggests a more sophisticated time series analysis, including lagged correlations, as shown by Braswell et al (1997) in Science, analyzing these same NDVI data, globally, though with respect to T only. Other prior work, using fluxes rather than NDVI also supports some of the author's assertions,

for example "Therefore, short–term and long–term processes can be obscured by the dominant influence of the annual cycle" was shown rather dramatically by Braswell et al 2005. Both the Katul and Braswell (97) suggest that climate variation in prior years can influence the response in a current year, through carry over via carbon pools, water storage and nutrient cycles and so an approach only considering very short lags may incorrectly characterize climate sensitivity or "co-oscillation". Katul's argument, that slower dynamics in the response of eg soil moisture, leaf area, may rectify forcing onto longer time scales, preserving entropy by distributing it to longer time scales and potentially producing complex response, is not addressed in this analysis, though the work is mentioned. Since others using these data, and flux data, which may reflect similar underlying coupled mechanisms, it seems that this analysis is at least incomplete, or the authors have not explained why they think time-scale crossing correlations aren't affecting their results. My concerns may be addressed with some additional discussion of lagged dynamics, or they could motivate additional analysis.

---

## Author Comment (AC1) · 7 Dec 2019

**Response to reviewers for manuscript "Towards a global understanding of vegetation–climate dynamics at multiple time scales"**

**Anonymous Referee #1**

This paper analyses temporal variability in vegetation greenness (NDVI), air temperature (T), and precipitation (P) over broad range of time scales and over the entire terrestrial landmass at 0.5 degree resolution. The overall purpose of the paper is to identify vegetation-climate co-variations. I am impressed by the scope of the work presented in this paper and by the very detailed description of the data and methods. The use of the NDVI as a metric of vegetation greenness could be controversial but the authors gave a good argument in favor of their choice.

We thank the reviewer for the positive evaluation of our work, the careful review and constructive feedback. We provide our point-by-point response below.

1. The choice of 0.5 degree as the spatial resolution has not been explained, I suspect that it was needed for averaging purposes due to lapses in data coverage at the higher resolution. However, it is important to remember that such resolution amounts to about 55 km on the equator. This is a significant area over which to average NDVI, T, and P, especially in the mountainous and coastal areas where significant spatial variability of these quantities are expected at that spatial scale.

We acknowledge the limitation of working at 0.5° spatial and 15-daily temporal resolution, especially in terms of losing signatures of short-term and/or local extreme events. The chosen resolution results from a trade-off in climate and vegetation data: On the one hand, consistent long-term meteorological datasets have a coarse spatial resolution (0.5°), which requires spatial aggregation of NDVI for direct comparison. On the other hand, the long-term NDVI dataset has a limited temporal resolution (15-daily), which requires temporal aggregation of climate data. Aggregating to the coarsest common resolution ensures conservative estimates, whereas downsampling would artificially increase data variability which may cause spurious results.

To explain our choice of spatial resolution, we will add details about and elaborated the reasoning behind on the Data section.

Suggested manuscript changes:

Methods - page 3, section 2.1 ll. 73ff:

"A global gridded dataset of NDVI AVHRR was retrieved from the Global Inventory Monitoring and Modeling System (GIMMS, Pinzon & Tucker, 2014) at 15–daily temporal and 0.083° spatial resolutions (GIMMS NDVI v3.1). Original data was aggregated to 0.5° by taking the mean of corresponding 0.083° pixels to match the spatial resolution of climate data. Air temperature (Tair), from the European Centre for Medium-Range Weather Forecasts (ERA Interim v4, Dee *et al.*, 2011) and precipitation (Prec) from the Multi-Source Weighted-Ensemble Precipitation (MSWEP, Beck et al., 2019) were aggregated to match  temporal resolution of NDVI by summation (Prec) or averaging (Tair). *Spatial resolution of T air was preserved (0.5°), while MSWEP mean values were used for pixel resampling (0.083° to 0.5°). Spatial and temporal resolution were fixed based on the coarsest resolution among the input datasets to ensure conservative results.* The time period considered was from 1 January 1982 to 31 December 2015."

Study limitations (section 4.5) on page 17, ll 381ff: "*Our analyses are conducted at 0.5° spatial and 15-daily temporal resolution, which may obscure short-term and local vegetation-climate relations, and instead only provide average relationships of variables within each grid cell. Our analyses may thus not be representative in heterogeneous landscapes such as coastlines or mountains.* Regions standing out through heterogeneous patterns, such as the Amazon, should be further investigated *regionally* at *higher* temporal and spatial resolution *wherever consistent data streams permit this* to better understand local influence of climate, vegetation and topography on atmosphere-biosphere co-variation."

2. A decision to work on the lat-lon grid is also left without comments. Using the lat-lon grid for global studies is questionable even if it is common in the literature. This problem is multiplied by the use of large grid cells, the shape and areas of cells at different latitudes change significantly. Projection to other grids, more appropriate to global analysis, is possible. Authors should discuss their choice of using the lat-lon grid with large grid cells.

We are aware of this discussion, but don't see a feasible alternative for our approach; It would be interesting to consider other projections and potential differences arising from those, but we see this out of the scope of the current paper. In particular we work on a data model that has been implemented for lat-lon-grids (Mahecha *et al.*, in review) so that we can efficiently parallelize code etc. Yet, we believe that the projection is secondary for the global aggregate statistics (in Fig 2, Sup. Tables 2-3) as we consider the spatial representation of each pixel and since neighborhood relationships between pixels are not relevant to our approach. When using lat-lon projections, results can be corrected for latitudinal differences by weighting by grid cell size or the cosine of latitude. This is explained in line 124 "*To identify emerging features occurring at different latitudinal bands, mean values weighted by pixel area were calculated in the tropics ...*". Nevertheless, we realize that this has not been stated clearly enough in the paper, and will add a clarification in the manuscript.

Manuscript changes: section 2.2 on data processing, page 4, ll. 85: *"Analyses were performed on Lat-Lon grid due to software specifications and data considerations. In all spatial analyses on Lat-Lon grid, the difference in size of grid cells between high latitudes and the equator was accounted for through weighting values by grid cell size. Similarly, in all analyses that involved random sampling of data points, the sampling probability was proportional to grid cell size."*

3. The Fourier transform method of dividing signal into short-term, seasonal, and long-term is standard and effective. The global maps of NDVI, T, and P variance decomposed into three regimes of temporal variability (short, seasonal, and long) shows (Fig.1) mostly what I would expect with the exception of short-term variability of NDVI. I think it would be useful if the paper would mention that seasonal variability of NDVI and T is a "default" variability –something expected. Long term variability of these two variables is something that we can understand, but short-term variability is surprising, especially for the NDVI which should have more "inertia" than T. On the other hand, short-term variability of P is expected as we all know from experience, seasonal variability of P is also expected and long-term variability could be understood. More generally, the paper describes results in detail but lacks commentary in a spirit of what I outlined above. This is not that we don't have any expectations of how the results should look like, so it would be effective to stress the unexpected parts of the results.

We thank the reviewer for the constructive comment and agree that the unexpected part of the results deserve more attention. We will modify the manuscript accordingly throughout to stress the expected and new results. Some examples are listed below:

Results - page 6, on section 3.1 lines 159ff and 165ff respectively:
"Assessing the contribution of each time scale sub–signal to the signal variance at each grid cell, we find that for NDVI most of the temporal variability is *expectedly* captured by the seasonal cycle (71% of the global variance), ..."
"Similarly, T air is strongly dominated by seasonal oscillations in the extratropics above/below 23.5 ∘ N/S (94% and 90%, respectively, Sup. Table S2) *as would be expected*."

Results - page 8, section 3.2 Classification of co–oscillations regimes ll.181ff:
"Combining the maps for all three variables into a map of co–dominant "oscillation regimes" (Fig. 2, Sup. Table S3), we find that seasonal NDVI regimes co–occur predominantly with seasonal T air, and seasonal or short–term Prec regimes (blue regions). *Dominant seasonal cycles of NDVI and Tair, as well as fast oscillation regimes in Prec, are expected over large parts of the globe, which is reflected by the large extent of the AAS and AAA classes in this analysis. Beyond this expected, solar–cycle–induced behaviour, a number of differentiated oscillation classes stands out:* Short–term NDVI oscillations occur mainly in the South American and Asian tropics, ..."

4. Authors have designed a clever scheme to show "co-oscillation regimes." It classifies grid cells' into 11 classes based on co-occurrence of dominant temporal variability regime (S, A, or L) for each variable (NDVI, T, P). This yields a compact and easy to understand map (Fig. 2). Again, I argue that it would be useful to stress that AAS and AAA classes are expected by default – something that we would predict on the basis of prior knowledge without analysis presented in this paper. Thus focus should be on the remaining classes. The LAS class can be easily explained, these are semi-desert or shrubland areas with only slightly seasonal-dependent vegetation and sporadic rainfall. On the other hand SSS, SAA, and SAS classes are difficult to understand. Authors do not try to understand them; instead they just say that this points to a complex climate-vegetation dynamics. I think that this is a way of saying "we tried to understand them but failed" or "we did not try to understand them in this paper". Both statements are acceptable but should be stated clearly. To me these classes are difficult to accept, for example, what does that means that some regions in Indonesia are characterized by predominantly short-term variability of vegetation and temperature? It seems counter-intuitive and contrary to prior knowledge. Could it be a data artifact, or method artifact? One possible explanation is that the short-term temporal variability include all periods < 0.9 year, so maybe in these regions the actual variability is ~0.9 year which would be more intuitive.

We thank the reviewer for the detailed discussion on Fig. 2 and agree that classes dominated by short-term NDVI require further discussion. Regions where short-term NDVI dominates are found mainly around the equator, where seasonal/annual cycles contribute little to overall signal variance. In southeast Asia and tropical South America, for example, they may be linked to the Madden-Julian oscillation (MJO; Madden and Julian, 1971) which has an oscillatory period between 20 and 90 days depending on the phase and region. MJO is defined by anomalies in the atmospheric pressure between 10°N and 10°S in the Indian Ocean region that propagates eastward to the Eastern Pacific (Madden and Julian, 1971, 1972). It is considered the dominant component of intra-seasonal variability in the tropics (Zhang 2012) and may cause short-term NDVI oscillations through recurring waves of precipitation where local conditions are affected by MJO. MJO is one feasible link of short-term vegetation responses to climate in the tropics, yet we are aware that short-term oscillations of vegetation in those regions need to be further investigated. Analysis at higher spatial resolution might reveal other sources of short-term variability related to the local climatic conditions as well as non-climatic processes such as land use change or crop rotations, among others.

As correctly described by the reviewer, periods <0.9 years are classified in the short-term bin except for the semiannual and 4-monthly harmonics (see methods). In fact, the cutoff between seasonal and short-term band is set exactly at 0.9 years, so signals with period of 0.9 years and above are classified as seasonal. Some of the overall short-term oscillations can thus be attributed to ~0.9 year periods, but this temporal class also includes oscillations with higher frequencies, such as the oscillations between 20-90 days mentioned above. A finer separation of frequency bins could aid in differentiating these patterns, but should be combined with an

analysis at higher spatial and temporal resolution in the regions of interest to better account for local and short-term effects.

Suggested manuscript changes:

Discussion, section 4.1 on page 14 with the following statements (ll.298):
"In contrast, heterogeneous spatial patterns are observed in equatorial and semi-arid regions, where different dominant scales of oscillation are found for NDVI and climatic variables. Here, the relationship between variables may depend on additional factors, and/or scales may show interactive effects. In the tropics, radiation is proposed to be one of the main drivers of NDVI (Nemani et al 2003, Seddon et al 2016), which could partially explain the lack of temporal coherence between NDVI, Tair and Prec. *Dominant short-term oscillations of NDVI (SSS, SAS, SAA) might be explained by climate intra-seasonality in the tropics due to the Madden Julian Oscillation (MJO). The MJO is defined as anomalies in the atmospheric pressure between 10°N and 10°S in the Indian Ocean region that propagates eastward to the Eastern Pacific (Madden and Julian 1971, 1972). Depending on the region and phase, its oscillatory period ranges between 20 and 90 days. MJO is considered the dominant component of intra-seasonal climate variability in the tropics (Zhang et al., 2013). We see MJO as one feasible driver of short-term NDVI oscillations through alterations of precipitation and temperature (Zhang et al., 2013, Hidayat et al., 2016, Mayta et al., 2018). However, MJO impacts, teleconnections and predictability are still insufficiently understood (Zhang et al., 2013, Wang et al 2019). Short-term oscillations of vegetation in those regions need to be further investigated; including other sources of intra-seasonal variation, connections with climate events, and data constraints. Additionally, regional analysis at higher spatial resolution might reveal details in local climatic variability, and other non-climatic processes such as land use change or crop rotations, among others.* "

5. Authors compare their map of "co-oscillation regimes" with the map of land cover (GLC) and, separately, with the map of climate (Koppen-Geiger or KG). The reason for such comparison is not clearly explained. On one hand authors describe their classification as new and different (from climate and land cover classifications), but, on the other hand, they look for similarities with those maps. I am not saying that such comparison is uninteresting, just that the rationale is not explicitly given. Also, maps could be compared using methods especially designed for such purpose (see International J. of GIS, 2018, v32(12), pp. 2386-2401).

Our goal in comparing different land cover classification schemes is to find the added value of our classification compared to "static" land cover and climate classifications, e. g. which KG or LC class shows the highest heterogeneity in terms of our classification (e. g. tropical evergreen forest), and which of them overlap well with an oscillation regime (e. g. KG class D and AAF).

This is possible with a pixel-by-pixel comparison, yet we agree with the reviewer on using a more advanced method such as the V-measure that complements our results. This index brings an overall perspective of the map associations based on homogeneity and complementarity criteria. We will introduce the V-measure outcomes included this new analysis in the manuscript and explain the motivation of our analyses.

Suggested manuscript changes:

- Methods - page 4, section 2.4 Variance per time scale and co-oscillation regimes, line 110ff: *"For the same area, we calculated the V-measure (V), a spatial association index based on homogeneity and complementarity criteria proposed specifically for thematic map comparison (Nowosad and Stepinski 2018). This measure provides overall comparisons of the spatial association between the co-oscillation regime map and Köppen–Geiger and GLC2000, respectively. Its range is from 0 to 1, with 1 being a perfect association."*

- Results - page 8, section 3.2 Classification of co–oscillations regimes, lines 200ff. " *We investigated to what extent our classification into* "oscillation regimes" *shows patterns of temporal vegetation–climate relations that are not represented by conventional classifications of the land surface. To determine overlap and differences between the classification of temporal vegetation–climate co–oscillations with static classifications of land cover (GLC2000) and Köppen–Geiger climate classes we assessed their spatial association with the co-oscillation regimes. The V-measures of co-oscillation regimes with Köppen–Geiger and GLC2000 were V = 0.17 and V = 0.11, respectively. These values indicate weak associations. However, we observe a stronger association with Köppen–Geiger than with GLC2000. Also when comparing homogeneity and complementarity (Sup. Table 4.), the co-oscillation regimes were more homogeneous with respect to climate than to land cover classes.* "

- Discussion - page 14, subsection 3.1 Comparison across time scales, line 290ff:
  "The classification provides an additional layer of ecosystem characterization beyond common classifications such as land cover classes or the effective Köppen–Geiger climate classifications (Geiger, 1954; Koeppen, 1900; Kottek et al., 2006) which only consider seasonality besides mean climate states, increasing our understanding of dynamic vegetation properties across time scales. *The complementarity of this data-driven classification of vegetation dynamics, extracted from the time series and summarized in the co-oscillation classification, is supported by the low spatial association calculated from the V-measure. Our findings show*  that the dominant oscillation of NDVI is often, but not always, related to dominant oscillations of T air and Prec (Fig. 2)"

- *Supplementary table 4:*

*Table S4. Spatial association between the co-oscillation regime map and Köppen–Geiger and Global Land Cover maps. c = complementarity, h = homogeneity, m = number of classes , V = V-measure.*

| | Co-oscillation regimes (11 classes) | | | |
|---|---|---|---|---|
| *Static maps* | *m* | *h* | *c* | *V* |
| *Köppen–Geiger* | *4* | *0.19* | *0.16* | *0.17* |
| *Global land cover* | *9* | *0.16* | *0.09* | *0.11* |

6. Authors also constructed maps (Fig. 3) on which a color (organized into a bivariate table) indicates correlation between NDVI and T signals, and, at the same time, a correlation between NDVI and P. As in the previous two figures, I found that the text should stress that seasonal cycle is a default and short and long term cycles are the "new" results. In this case, however, because of the uniqueness of the bivariate map, even the seasonal map is new and could use more explanation going beyond just describing what we already see in the map.

We will rephrase a paragraph in section 3.4, page 10, ll. 230ff to reflect the suggested changes as follows: "On the seasonal scale, NDVI correlates positively with Tair and Prec above 40°N, whereas at the other latitudes all possible relations are observed. *Especially South America shows a highly diverse pattern of correlations. Differences exists across the tropics, where South America and South–East Asia display mainly negative correlation with Prec, while the African tropics display positive correlations with Prec. Semi–arid regions show negative correlations with Tair as would be expected. While some of the patterns are known, this* correlation of decomposed oscillations reveals a more differentiated picture of ecosystem variability in comparison with the undecomposed data (Sup. Fig. S7). *Notably, correlations on short– and longer–term scales partially show opposite signs compared to the seasonal scale, e. g. in South America, South Africa and Central America.*"

7. Overall, I find this paper to be solid, very interesting and illuminating; however, it is, to my taste, overly skewed toward technical description at the cost of not providing plausible explanations to some of the more unexpected results.

We appreciate the constructive comment and agree with the reviewer on the need to state the unexpected rather than the obvious. We suggest to modify the manuscript throughout to account for this suggestion, as outlined above.

**Anonymous Referee #2**

This is a nice analysis, although the significance for ecosystems and climate is not stated clearly anywhere in simple language.

We thank the reviewer for the overall positive evaluation of our work and the important points raised. We provide our point-by-point response below.

In contrast to other comments, this is not the first such analysis (see for example Braswell et al 1997 and Martinez and Gilabert 2009) as well as substantial other work on time scales using related data such as CO2 (Vukicevic et al 1997). This analysis, while acknowledging earlier work, doesn't build very much on insights from earlier work though it uses longer and more robust time series.

Thank you for this comment. While individual papers may not have been given sufficient attention, we respectfully disagree with the perception that we did not build on the "insights from earlier work". Rather, our study was motivated by the site level and regional studies that were mentioned here. Differences setting our study apart from previous studies are the assessed temporal and spatial scales, use of satellite data versus direct measurements, and global spatial coverage, which address new challenges. For example, analysis derived from flux towers as Katul et al (2001) or Braswell (2005) is representative for a single, homogeneous land cover (i.e. pine forest) and does not need to consider spatial variation. However, applying time series decomposition in the spatial domain turns into a challenging factor for selecting methods with proper binning to be spatially consistent. Martinez and Gilabert 2009 use wavelet decomposition for Spain, and Hawinkel et al. (2015) use EEMD in East Africa, both methods being applied successfully at regional scales. But spatial heterogeneity turns out to be limiting when contrasting areas such as the tropics and temperate zones are treated uniformly. Another difference is our data-driven approach and the characteristic of satellite-derived data; as a consequence, the analysis of fluxes and models for understanding changes in pools (as in Vukicevic *et al.*,2001, and Katul *et al.*, 2001) is out of our scope here. Nevertheless, we do see the point that these papers have not been addressed in sufficient detail - in the revised version of the paper we will put more emphasis on the known patterns as reported by Braswell, Katul and other leading scientists in the field.

Specifically, Katul's analysis (2001, I think) that showed that ecosystems may rectify short-term forcing onto longer time scales suggests a more sophisticated time series analysis, including lagged correlations, as shown by Braswell et al (1997) in Science, analyzing these same NDVI data, globally, though with respect to T only.

An important point raised here is the consideration of lagged vegetation responses. This question - addressed at the global scale - is not trivial. Several studies have shown different lags for different vegetation types and among regions. This complexity even increases when different time-scales are considered. Here, lags may be of the order of days to weeks for short-term oscillations, and weeks to months on a seasonal scale. Especially for the longer-term scale, time lags from months to years may co-exist depending on the processes at work and non-linear interrelations between different variables. It should be noted than Braswell *et al.* (1997) cross-correlated monthly NDVI anomalies and $CO_2$ growth rate with mean temperature. The equivalent of monthly anomalies in our study are signal fluctuations distributed between different time scale bands, depending on their frequency. Observed lag effects would thus not be directly comparable (or may have alternative explanations, see Wu *et al.*, 2008), but it is perceivable that lagged effects of several years could exist on the longer-term anomaly component.

We agree that global patterns in time lag analysis may bring additional valuable insights into ecosystem functioning and should be studied. Clearly, lag analysis considering autocorrelation and multivariate partial correlations at multiple time scales is not a trivial undertaking. Comprehensively assessing lagged relationships across time scales and determining in how far they change with time scale would be a rather complex analysis, if not a paper in itself, which in our opinion exceeds the scope of the current paper. For the current study, we thus sought to bring forward a first global understanding of vegetation response to climate across time scales, based on one consistent time lag for simplicity. Recent studies assessing time lags between vegetation and climate indicate that time lags of around one month carry most explanatory power for predicting vegetation dynamics at comparable temporal and spatial resolutions (Kraft *et al.,* 2019, Krich *et al.*, in revision). We did perform initial analyses to consider which time lag should be chosen for the correlation analysis beyond findings from literature. A lag of one time step in our study is based on this previous data exploration that was not included in the paper, but reported later to the editor. Due to the 15-daily temporal resolution of the data, a response time of up to 15 days is intrinsically included in our analyses. Each time lag is therefore an additional 15 days, and shorter responses cannot be assessed. We compared 6 different lags (from 15 to 90 days). When correlating NDVI and precipitation instantaneously, we found almost exclusively negative correlations for the short-term scale. A lag of one time step was sufficient to arrive at more expected positive correlations between NDVI and precipitation, while lagging NDVI further behind precipitation did not substantially improve or alter the results. We thus chose to globally use the lag of one time step (representing 15-30 days response time) between precipitation and NDVI across all scales. Globally, temperature appeared to be most strongly correlated to NDVI instantaneously, thus no time lag was introduced between air temperature and NDVI.

In the revised manuscript, we will be happy to include an ancillary figure (see below) on this time lag analysis and discuss their possible influence, as well as corresponding limitations of our approach more thoroughly, as outlined above.

Suggested manuscript changes:

- Add an explanation of the choice of time lag in the methods, section 2.5 Correlation between variables at each time scale, page 5, ll125ff: "*Due to the 15-daily temporal resolution of the data, a response time of up to 15 days is intrinsically included in our analyses. Each time lag is therefore an additional 15 days, and shorter responses cannot be assessed. We compared 6 different lags (from 15 to 90 days). When correlating NDVI and precipitation instantaneously, we found almost exclusively negative correlations for the short-term scale. A lag of one time step was sufficient to arrive at expected positive correlations between NDVI and precipitation, while increasing the lag time did not substantially improve or alter the results. We thus chose to globally use a lag of one time step (representing 15-30 days response time) between precipitation and NDVI across all scales. Globally, temperature appeared to be most strongly correlated to NDVI instantaneously (not lagged), thus no time lag was introduced between air temperature and NDVI. Recent studies assessing time lags and memory effects between vegetation and climate also indicate that time lags of around one month generally carry most explanatory power for predicting vegetation dynamics (Papaginnopolou et al., 2017, Kraft et al., 2019, Krich et al., in revision).*"

- add a discussion to the limitations, section 4.5, page 17, ll. 377ff: "Analysis of time lag effects between atmospheric forcing and vegetation response may bring additional valuable insights into ecosystem functioning, yet assessing meaningful time lags across time scales is challenging due to a variety of  processes *involved* . *Plausible time lags from months to years have been suggested between climate forcing and vegetation response and/or ecosystem carbon exchange through direct and indirect effects (e. g. Braswell et al., 1997, Braswell et al., 2005, Vuikchevic et al., 2001, Krich et al., in revision, Kraft et al., 2019, Papaginnopolou et al., 2017). Assessing lagged vegetation responses across time scales may help to disentangle such co-existing time lags to form a global, time-scale-resolved picture of vegetation responses to climate. To account for the confounding effect of autocorrelation and spurious links between variables, methods like causal inference (Runge2014, Runge2019, Krich et al., in revision) should be applied in order to retrieve causal time lags between variables.*"

[Figure]

**Figure: Lag of maximum absolute correlation between NDVI and air temperature (T) at each grid cell  (left panel) and NDVI and precipitation (Prec, right panel).** The time-step used is 15 days, which is equivalent to a 0.5-month lag in the color key.

Other prior work, using fluxes rather than NDVI also supports some of the author's assertions, for example "Therefore, short–term and long–term processes can be obscured by the dominant influence of the annual cycle" was shown rather dramatically by Braswell et al 2005. Both the Katul and Braswell (97) suggest that climate variation in prior years can influence the response in a current year, through carry over via carbon pools, water storage and nutrient cycles and so an approach only considering very short lags may incorrectly characterize climate sensitivity or "co-oscillation". Katul's argument, that slower dynamics in the response of eg soil moisture, leaf area, may rectify forcing onto longer time scales, preserving entropy by distributing it to longer time scales and potentially producing complex response, is not addressed in this analysis, though the work is mentioned.

Since others using these data, and flux data, which may reflect similar underlying coupled mechanisms, it seems that this analysis is at least incomplete, or the authors have not explained why they think time-scale crossing correlations aren't affecting their results.

My concerns may be addressed with some additional discussion of lagged dynamics, or they could motivate additional analysis.

We thank the reviewer for this comment and will modify the manuscript to discuss the points mentioned. In addition to the points stated above, carryover effects from previous years may not be detectable when only assessing NDVI as response variable, and at the given coarse spatio-temporal scale. Additionally, the impacts of a given extreme climatic condition on the carbon cycle may differ on direct versus indirect, instantaneous versus lagged, and local versus global scales (Frank *et al.*, 2015). Most references discussed assess flux data or additionally consider $CO_2$ dynamics, and/or a modeling framework, to conclude on direct, indirect and carry-over effects. While direct impacts of extreme events can be well captured at specific sites, such as Eddy Covariance towers, it is much more challenging to attribute direct and indirect effects of extreme climate on the coarse scale of remote sensing products like GIMMS NDVI, especially when other important drivers such as cloudiness or fires are not explicitly assessed. We will modify the manuscript to highlight these limitations.

Suggested manuscript changes:

In addition to the suggested manuscript in section 2.5 (page 5, ll125ff) and section 4.5 (page 17, ll. 377ff) introduced in the previous point response, we will add to the discussion, section 4.2, page 15, ll. 324ff: " [...] generating a possible link between short-term and longer-term scales. *Previous studies suggest that climate forcing on one time scale can be amplified or dampened in corresponding vegetation responses (Stoy et al., 2009), or transferred to another time scale (Katul et al., 2001), preserving the system's entropy but creating complex interactions across scales. This highlights* the need to further investigate interactions between different time scales globally in long-term EO records."

**Unsolicited review #1**

In the paper Towards a global understanding of vegetation-climate dynamics at multiple time scales, Linscheid et al. describe a novel approach to study the dynamics between vegetation and climate at multiple time scales. Due to long-term Earth observations (EOs), ecosystem analyses are possible at time scales of over thirty years. In this study, global NDVI is correlated with two climate variables, air temperature (Tair) and precipitation (Prec), and studied at short-term, seasonal, and long-term time scales. The different time scales were determined using Fast Fourier Transformation (FFT), and linear correlation methods were used for the correlation. Bivariate colour maps were used to present the correlations. The effect of land use change was also taken into account, to see its effects. The authors did a comparison between the techniques Fourier Transform and Empirical Mode Decomposition, to see whether the latter more data-adaptive technique leads to different insights.

The paper was written clearly and is well-structured, and the figures were clear and supportive of the paper. Using bivariate colour maps to present the correlations is an excellent choice for the variables considered in this research (Teuling et al., 2011). Similar research has been done before, but only for specific regions (Martinéz and Gilabert, 2009; Canisius et al., 2007; Hawinkel et al., 2015) or not taking co-interpretation with climate variables into account (Pan et al., 2018), which is why the findings in this paper are novel and valuable. The paper fits the scope of the journal Biogeosciences very well, but some revision needs to be done before the paper is ready for publication. The paper overall is good, but revisions could be made on several topics. I recommend the authors to revise some parts of the paper. Specifically, in sections 2.1, 2.2, 2.3, 2.4 and 2.5 I have some major revisions. I also have some further minor and very minor revisions. I will go into further details on those revisions below.

We thank the unsolicited reviewer for the time spent on a thorough and detailed review, and the positive evaluation of our work.

Major comments:
1. In section 2.5 the method for the correlations between variables at each time scale is discussed. Correlation of the time-scale specific sub-signals of NDVI, Tair and Prec was done using Pearson's correlation coefficient, Spearman correlation, and partial correlation. All three of those methods assume linear correlation. However, the authors do not specify anywhere in the paper why, or even if, linear correlation can be assumed for the data they are analysing. This assumption should be elaborated on, because if linear correlation could not be assumed, these correlation analyses would not be a possible method. The choice of linear correlation affects results section 3.4 and Figures 3 and 4, thus revision is needed. My proposed revision entails the authors explaining which assumptions are made to allow for linear correlation. It is also advisable to discuss the consequence of assuming linear correlation in the discussion, to evaluate its effect on the results of the study.

We acknowledge the limitation of working with linear metrics to characterize an often non-linearly responding bisophere and climate system. However, non-linear methods do not yield a directionality of variable relation, e. g. corresponding to a positive versus negative correlation, that could have different functional interpretations. Differentiating positive from negative relations was important in this study, and we thus stuck to linear analyses. We included Spearman correlation as this method does not require a linear, but only monotonic relationship between variables. Nevertheless we acknowledge that our approach is a first exploratory analysis, and more robust methods, including non-linear analyses, should be considered in future studies to capture additional dynamics of vegetation-climate interaction.

2. In section 2.1 the data that is used for the study is elaborated on. A global gridded dataset of NDVI at 0.5° spatial resolution is used. Tair and Prec data is obtained at 0.083° spatial resolution. To match the NDVI data, the data for Tair and Prec were aggregated to 0.5° spatial resolution. For Tair this was done by averaging, and for Prec this was done by summation. However, detail on extreme values for both Tair and Prec is lost by this aggregation. For Tair, local extreme values will be lost due to averaging over a larger area, while for Prec local extreme values will be smoothened out due to summation over a larger area. Upscaling of the T air and Prec data is part of the pre-processing of the data, thus it has effects on all results and conclusions. A revision is recommended to include a quantification of the sensitivity of the resulting classification in regimes to the spatial averaging.

   Furthermore, NDVI data is available at 0.05° spatial resolution (National Center for Atmospheric Research Staff), so the upscaling of the Tair and Prec data to 0.5° is not necessary. There could be many reasons for not choosing the higher resolution 0.05° data, for instance due to the 0.5° data being more manageable. However, there is no discussion on these possible reasons. Revision is therefore advised to include a discussion about the choice for 0.5° rather than 0.05° spatial resolution data.

Please see answer to reviewer 1, point 1.

3. In section 2.2 the time periods that are used to reconstruct the different time scales are described, stating that similar frequency bins have been used in previous studies. The literature referenced however, by Mahecha et al., has no mention of these similar frequency bins. The article only states that the annual-seasonal was defined as periods in the interval 0.375-1.25 years. I could not obtain the other literature that was referenced for these similar frequency bins, a thesis by Fürst. Basing the time periods only on the annual-seasonal interval mentioned in the article by Mahecha et al. without further explanation is not valid. This section should be revised, and argumentation for the chosen time periods should be included, as well as a quantification on how the conclusions are influenced by these time period boundaries.

Mahecha *et al.* make use of three frequency classes, which they explain as follows *(p. 2311, section 2.2)*: "*The choice of frequency classes is subjective and constrained by the length of the FAPAR series, its temporal resolution, and the notoriously prominent annual cycle. For simplicity, we use only three intuitive frequency classes: The annual–seasonal scale, framed by two bands of lower and higher frequencies, respectively. The annual–seasonal bin is defined by periods in the interval {0.375; 1.25} years.*" Hence their "higher frequency" band contains all periods <0.375 years and their "lower frequency" band all periods >1.25 years, although admittedly this is not explicitly stated. Our approach employs binning of sub-signals into three corresponding frequency classes, although the exact limits differ slightly due to intrinsic differences in SSA and FFT, whose results yield sub-signals of different frequencies.

We will extend our explanation for the frequency binning and clarify the citation in Section 2.3 Time series decomposition, page 4, ll. 94ff:
"*The sub-signal binning was centered on the definition of the seasonal/annual bin similarly to Mahecha et al. (2010a) and Fürst (2009). The bin ranges were slightly adapted due to the FFT approach, which yields signals of different frequencies compared to the approach chosen by Mahecha et al. (2010a).*  To identify emerging features occurring at different latitudinal bands…"

4. The time series decomposition is done by Fast Fourier Transformation. However, Mahecha et al. (2010) state that classical Fourier decomposition is not the best method to segregate data into different temporal scales. Discrete Wavelet Transforms (DWT) or Empirical Mode Decomposition (EMD) are mentioned as better alternatives, because those methods do not assume a fixed superposition of weighted sines and cosines. The authors do a comparison of Fourier Transform with EMD in section 2.7 of the methods, and they discuss the difference in results between the two techniques in section 4.4, but it is recommended to include an explanation on the choice for FFT as method for the main research.

The reviewer correctly criticizes FFT for not being the optimal decomposition method, mainly based on the fact that FFT assumes stationarity, fixed frequency and fixed amplitude of a given sub-signal. Its advantage however is much greater computational speed in comparison with CEEMD, which is an iteration-based ensemble approach. The binning approach taken in our study alleviates part of the aforementioned limitations since single time-invariant FFT signals at each frequency are combined into three frequency classes which *de facto* yield non-stationary, amplitude- and frequency-modulated signals for each time scale considered. We further show that these FFT-based sub-signals yield very similar results to the CEEMD-based approach in an example case over Europe (Sup. Figs. 13-16), justifying the use of FFT in this particular case where it is combined with a frequency-binning approach. Applying CEEMD globally is certainly desirable in our eyes, yet we have found that CEEMD results are spatially heterogeneous and especially sensitive to the noise parameter in case of CEEMDAN (unpublished). Given the

remarkably similar results between CEEMD and FFT over Europe, we thus currently see FFT as a justified compromise. We will clarify this position in the revised manuscript.

5. In section 2.5, lines 124-125, it is stated that NDVI was lagged one time step (15 days) behind Prec in order to allow response time of vegetation to changes in water availability. However, Jamali et al. (2011) found a response time of 8-40 days of vegetation indices to rainfall changes for six sites in Africa. This response time is much longer than the 15 days assumed in this paper. The study by Jamali et al. was conducted in Africa, so perhaps in more moderate climates, the response time would be closer to 15 days. A revision is recommended to include a motivation for the 15 days response time chosen for this study, to discuss on what literature this choice was based and whether this is a valid choice for a global estimate.

Please see our answer to referee 2 dealing with lagged responses.

6. In section 2.4 the method for determining the variance per time scale and co-oscillation regime is discussed. It is explained that 64 possible combinations of oscillation regimes are possible, of which only 26 occurred. For simplicity, the analysis focussed only on the 11 most abundant oscillation regimes, which comprise 99.7% of pixels. The only argumentation given by the authors for choosing these 11 most abundant oscillations is for simplicity, which is not a concrete reason to make this choice. The choice to take only the 11 most abundant oscillation regimes into account affects results sections 3.2 and 3.3, as well as Figures 2 and 3. I therefore recommend the following revisions: the authors should elaborate on what they mean by simplicity. They should also specifically discuss why the eleven most abundant oscillation regimes are chosen, rather than a more logical and round number such as 10 or 15. The same goes for the percentage of pixels, why choose 99.7% rather than for example 99.5% or 99%?

For selecting the number of classes we applied a threshold of 50 pixels (~150.000 km$^2$) resulting in 11 classes. The remaining classes were reclassified as "others". We show a total of 12 classes in Figure 2a, of which the first 11 classes are highly representative of the dataset (> 99.5%). The remaining categories accounted for regions too small to be individually distinguishable.

Minor comments:
No information is provided on data processing of NDVI to minimise effects by external factors. Some external factors that could have an effect are mentioned by Zeng et al. (2013): solar zenith angle or volcanic stratospheric aerosol effects from major volcanic eruptions. Please include a discussion on possible effects by external factors.

p5, lines 126-128: only the seasonal and longer-term oscillations are compared, and **no** motivation is provided for not including short-term oscillations in this comparison. Please elaborate on why short-term oscillations are not included in the comparison.

p9, Fig 2a: it is not specified what the white colours on the map correspond with. It can be assumed that these correspond with NA, but this should be mentioned in the caption.

p12, Fig. 3a: just as for Fig. 2a, no specification on what the white colours on the map correspond with. The caption mentions that NDVI<0.2 were excluded, but it is not mentioned that these are shown in white on the map.

p13, Fig. 4: same as for Fig. 2a and Fig. 3a. The caption says that areas with correlations between -0.2 and 0.2 were not considered, but it is not mentioned that these are shown in white on the map.

Very minor comments:
p4, line 110: change notation from "75'871'486 km2" to "75,871,486 km2".
p7, Fig. 1: in the caption, change "seasonal(annual)" to "seasonal (annual)".
p9, Fig. 2: in the caption, "DBF" is mentioned, but in the figure it is called "DBF_open". Please change one of the two, so they correspond with each other.
p10, line 219: change "At" to "at", because after a colon.
p10, line 228: change "While" to "while", because after a colon.
p10, line 237: change "For" to "for", because after a colon.
p10, line 239: change "northern" to "Northern".
p11, line 269: change "Tair" to "Tair".
p11, line 270: add a comma between "method" and "a".
p12, Fig. 3: remove the second "." in the bold part of the caption.
p13, Fig. 4: change "Tair" to "Tair" in the legend of the bottom map.
p14, line 274: change "characterised" to "characterized" for consistency, because in the rest of the paper American spelling is used.
p14, line 285: change "point" to "points", because it is a singular verb.
p14, line 293: add a comma between "precipitation" and "are".
p15, line 308: change notation from "3'939'362 km2" to "3,939,362 km2".
p18, line 400: change "Tair" to "Tair".

We thank the reviewer for the detailed review and comments and will clarify / correct the text when appropriate.

**Unsolicited review #2**

Disclaimer: This review was written by MSc student Le Lu as part of his course work on "scientific reviewing", under supervision of dr Arnold Moene from Wageningen University. The comments were submitted because they can contribute to the scientific process, and because they contain helpful questions and suggestions for the authors. Although the structure of this review follows the formal conventions, it is thus not a solicited peer-review from the editor of ACPD.

**Review of Towards a global understanding of vegetation–climate dynamics at multiple time scales** by *Linscheid et al.* by reviewer Le Lu

Linscheid et al. provide a global assessment of vegetation-climate variability at multiple time scales. Motivated by the existing knowledge gap regarding how the global terrestrial biosphere responds to multi-timescale variability of climate, the study aims to explicitly examine the variability of biosphere and climate time series, explore spatial variability of vegetation-climate dynamics at each time scale, assess the potential link between the resolved temporal patterns and traditional land cover classification and compare correlations of climate and vegetation across multiple time scales.By decomposing over 30 years of remote sensing records of NDVI and climate time series with Fast Fourier Transformation (FFT), variances in each variable contributed by different time scales from short-term oscillations to seasonal and long-term trend were compared and co-oscillations between NDVI and climate variables (i.e. temperature and precipitation) were discussed. Moreover, the oscillations regimes were compared with traditional classifications of land covers and climate zones.

For every time scale, the researchers also examined the correlations of NDVI with climate variables. Finally, the potential effects of land cover change, limitations of FFT and NDVI on time series decomposition were also considered.The results show heterogenous response by vegetation to climate variables both temporally and spatially. Section 3.1 presents that NDVI and temperature variability is dominated by seasonal cycle whereas 52% of the variance in precipitation is contributed by short-term oscillations on a global scale. However, differences exist as the long-term oscillation in NDVI is strongly associated with shrub and herbaceous land cover types. In section 3.4, a comparison of correlations of NDVI with climate variables suggests that correlations strongly vary with time scales and space. In South Africa, correlation with temperature is negative on short-term scale and long-term scale but positive on seasonal scale. Different correlations patterns are observed among tropical forest regions. Section 4.4 concludes that the results are proven to be robust by repeating the analysis with Enhanced Vegetation Index (EVI) and NDVI from the Moderate Resolution Imaging Spectroradiometer (MODIS) as indicators of vegetation dynamics or Empirical Mode Decomposition as the time series decomposition technique. The study is, as its authors claim, the first of its kind. It is the

first attempt to examining the global vegetation response to climate across multiple time scales. What is particularly strong about this study is that its methodology takes into account various sources of possible confounding factors that may compromise the results. As mentioned above, time series decomposition is repeated with a more data-adaptive approach. Considering the noise introduced by cloud cover, which NDVI is subject to, vegetation indices from MODIS are included as part of the repeated analysis so the results can be proven robust. The effect of land use and land use change (LULUC) is evaluated as well given the large time span that the study incorporates. Last but not least, the presentations of results are clear and easy to follow, particularly figure 2a in which the dominant oscillations of NDVI, temperature, and precipitation are indicated by a three-letter scheme.

Overall, the study is well-thought-out and well written. This is a study of biosphere-atmosphere interactions, so it falls well within the scope of the journal. It serves as an important contribution to our understanding of past and current climate-vegetation interactions on separate times scales, and its findings may even help us project the future interactions. In the meantime, however, there are a few issues that I believe require further attention and revisions before publication. These issues are discussed below in detail.

We thank the unsolicited reviewer for the time spent on a thorough and detailed review, and the positive evaluation of our work.

**Major arguments**

**1)** The methodology of this study relies on the use of long-term remote sensing records of NDVI. It is good to see some limitations of remote sensing NDVI are addressed by comparing the results with alternative data source and vegetation index such as EVI and NDVI from MODIS. Yet other known issues of NDVI are not discussed in the manuscript. It is well documented that NDVI can be biased in regions where biomass is dense (Pettorelli et al., 2005). NDVI saturates as plants grow and thus is insensitive to dense biomass (Huete et al., 2002). the saturation of NDVI may cause underestimated seasonal cycles in tropics, where vegetation is typically dense. This potential outcome could be an alternative explanation to the contradiction between the results of this study and Huang et al. (2019) regarding the negative correlations with temperature in the tropics. Furthermore, a recent study, which is cited for other parts of the study, attributes the low response of vegetation to precipitation in African tropics as partly a result of saturation (Hawinkel et al., 2015), contradicting the claim that the correlation between NDVI and precipitation is almost always positive. In another previous study on global NDVI time series, tropical evergreen forest is even excluded because of the low signal/noise ratio associated with saturation (de Jong, Verbesselt, Schaepman, & de Bruin, 2011).

Besides, NDVI is also influenced by reflectance from soil when the Leaf Area Index (LAI) is under 3 (Pettorelli et al., 2005; Huete, 1988). Specifically, darker soil substrates would result in higher values of NDVI for a given amount of vegetation. Huete (1988) notes that the NDVI suffers from soil reflectance issues across different soil types especially in global data sets. This

is critical to this study and likely to have implications on its results since the global scale is exactly what it deals with and involves naturally various soil types. For example, the results of the study show strong associations between the long-term NDVI regimes and arid regions where vegetation is sparse, but this could be partly the result of varying brightness of soil controlled by soil moisture. Meanwhile, the short and long-term correlations with precipitation presented by the study may be overestimated due to the higher NDVI values associated with darker soil substrates led by precipitation. These unaccounted impacts on the current results, without further looking at the pattern of variations in soil brightness, is unpredictable. Nonetheless, it would either overestimate or underestimate the contribution of oscillations at certain time scales to the overall variability considering the influence of noise introduced by soil background on NDVI.

To address the above limitations of NDVI, I recommend validating the results by repeating the analysis with another two vegetation indices just as what was done with MODIS EVI, at least for regions where vegetation is dense or sparse. The two indices are the Soil-Adjusted Vegetation Index developed by Huete (1988) and the Wide Dynamic Range Vegetation Index by Gitelson (2004), which are designed to overcome the noise from soil background and saturation of NDVI, respectively. Both indices are modified versions of NDVI and require only the reflectance inputs in the same waveband as what NDVI does, so there is no need to introduce any new data. Although the values of different indices are not directly comparable, which may compromise the global scale that the study aims at, in this way it nevertheless would be clear whether the two limitations of NDVI have any major implications on the results, providing a more accurate characterization of vegetation-climate dynamics.

We thank the reviewer for important concerns about NDVI as a proxy for vegetation activity. We are aware of the saturation effect in NDVI against GPP, as well as soil contamination of the signal, and agree that these limitations should be clearly stated in the revised manuscript.

In addition, we would like to highlight that NDVI was selected because it offered the longest record of vegetation greenness available, which was pivotal in our study. It is the most widely used vegetation index globally, therefore communicating its advantages and limitations allows a careful interpretation. In our study, we cannot conclude about photosynthesis or vegetation productivity directly, but seek to bring insights on which features of vegetation activity are abstracted at different time scales of such long term records. The GIMMS dataset was homogenized from different AVHRR instruments, corrected for view geometry and artifacts such as volcanic eruptions (Pinzon & Tucker, 2014). Unfortunately, this dataset does not offer the individual bands but only the computed NDVI. Therefore calculating the suggested indices would involve a different dataset. Nevertheless, it is relevant to mention that SAVI, the first suggested index, is closely related to EVI which partially reduces the influence of soil and canopy variations (Huete 1997, Huete et al 2002). A previous comparison between the shorter MODIS EVI and NDVI records yielded similar results (Figure S4). However, we do consider that soil reflectance might play a role in our analysis. In the future, we plan to assess other variables such as solar-induced fluorescence (SIF) which are more closely related to vegetation activity.

Discussion section, page 15 ll. 344ff: "Similarly, temperature is not usually limiting canopy development in the tropics (rather the contrary, Huang et al., 2019), which may explain negative correlations with Tair. *As NDVI saturates over regions of dense vegetation, results in the tropics need to be interpreted with caution, and negative correlation with Tair could alternatively be explained by underestimation of the seasonal cycle over tropical EBF*."

Limitations section, page 17, ll. 373ff: "*Known limitations of NDVI include saturation effect at high canopy cover, especially relevant in the tropics, as well as the influence of soil reflectance in sparsely vegetated areas. These effects could thus influence our results and* the emerging patterns should be compared with newer satellite products such as sun-induced fluorescence (SIF), which are coupled more directly to plant physiology and photosynthesis [...]"

**2)** The study attempts to assess the influence of land use and land use change (LULUC) on its reported NDVI classifications and concludes no widespread effect in section 3.3. The cutoff of land use change is set at pixels with 25% change in fraction of vegetations. There is no reason given regarding why 25% is chosen, which makes me curious to learn what the results would be if a threshold lower than 25% is used, for example, 20% or 15%. More pixels would be inspected with a lower threshold, and maybe more pixels would reflect change in NDVI due to LULUC. In addition, the study does admit the absence of marked signs of LULUC is perhaps because of the coarse spatial resolution of the data. Despite knowing the limitation, the study still concludes there is LULUC has no widespread effect on the results, which I find very puzzling. To me, given the limitation in spatial resolution the best that could be concluded would be "no definitive conclusion can be given" instead of lines 224 and 225.

As a side note, some questions regarding the validity of the study may be asked because the coarse resolution of the data is proposed as the explanation for the absence of signs of LULUC in NDVI. If this explanation holds, would it be suggesting that the spatial heterogeneity of vegetation dynamics is underestimated by the coarse resolution?

Regarding the assessment of the effect of LULUC, I recommend to either give reasons for the chosen 25%cutoff or compare the results with a lower threshold value. For the validity of the conclusion on LULUC effect and even the entire study, it is perhaps wise to recognize the coarse resolution as a limitation to the study design and be clear about possible implications. This addition could be put into section 4.5 Limitations and Outlook to give a more balanced reflection on the methods and results.

We assess pixels with >25% land use or land cover change (LULCC), which was the highest threshold applied and only retained few (<50) pixels per land cover class at the given resolution. We thus looked at the set of the pixels in the dataset which was *most affected* by LULCC. Even in this subset of pixels, we barely found any reflection of LULCC in the time series, and Sup.

Fig. S6 shows the few hand-picked examples which did show a clear LULCC progression. Lowering the threshold of LULCC was examined, but as expected, LULCC was even less visible in these sets since there was proportionally less of it. In ll. 224-225 we specify that "the change of vegetation over time did not have a widespread influence on the classification of dominant scale and oscillation regimes at the given spatial resolution", i. e. we specify that the two analyses which mainly rely on the variance of the sub-signals (Figs. 1+2) were not strongly affected by LULCC as we could not detect a strong shift in signal variance/amplitude even in strongly LULCC-affected pixels, which we think is a logical and plausible conclusion given that LULCC was largely not detectable in NDVI time series. We further state in ll. 219ff that "at half–degree resolution, most pixels represent mixed signals which obscure most of the details that would allow for detecting land cover changes", however, we agree that it is advisable to mention the limitation of working at a coarse spatial resolution in general. In concordance we have modified the section on study limitations, as discussed in our answer to reviewer 1, point 1.

**3)** Section 3.5 compares results from CEEDMAN with the FFT in order to prove the results are robust. However, hardly any justifications are given for carrying out this analysis other than lines 265 –266 "the data-adaptive empirical mode decomposition (EMD) could be expected to be better suited for exploring instationary ecological processes over time". It inevitably leaves readers to wonder about the relevance of this repetition, especially given that the results are largely the same. Moreover, the manuscript does not provide enough information about how the repetition was implemented. It recognizes the constraints posed by a higher spatial heterogeneity in implementing a global analysis with CEEDMAN, and it appears to allude the implementation includes a test case over Europe. However, it is not clear if any other regions were included as a part of the repeated implementation and if the results were based on any regions in addition to Europe.

I recommend the authors to be explicit in this section. Multiple papers are already cited regarding CEEDMAN, and I do not see why not convincing readers about the use of it by enumerate the benefits/advantages of this alternative method and the expected differences that the method may bring. It could also help readers better understand the actual yielded differences shown in lines268 to 270. It may also be useful to include more details as what the spatial scope of this repeated analysis instead of simply presenting the results of a test case over Europe.

The reviewer criticizes that we argue for the use of CEEMDAN, but our argument actually is made the other way around: data-adaptive methods like CEEMDAN are state of the art for time series decomposition, yet their global application is challenging due to the reasons mentioned in the manuscript and as explained in our answer to unsolicited reviewer 1, point 4. By comparing to a test case over Europe, we show that using Fourier decomposition in our case can be justified as it gives largely similar results to CEEMDAN. Please also see our answer to unsolicited reviewer 1, point 4.

**Minor issues**

1. In line 39 the authors state that only reflecting greenness is a limitation of NDVI, followed by multiple citations. After skimming the cited manuscripts, I found no relevant claims. One of the studies says the ability of NDVI to quantify greenness is what allows it to correlate with vegetation biomass and dynamics. In a rather thorough literature review I read regarding the use of NDVI for assessing vegetation response (Pettorelli et al., 2005), no relevant information is given about whether reflecting only greenness as a limitation of NDVI. It could be a factual error or the experience of the authors, but unfortunately no elaborations on this point are given. Please offer some evidence to support this claim.

Indeed, although some of the references summarize limitations of NDVI, they are inserted at the wrong position in the sentence, it should be clarified as follows: "*Vegetation indices such as the Normalized Difference Vegetation Index (NDVI) have often been interpreted as proxies for vegetation activity (Zeng et al., 2013; De Keersmaecker et al., 2015; Hawinkel et al., 2015; Kogan and Guo, 2017; Pan et al., 2018), despite well–known limitations of only reflecting vegetation greenness .*"
Only reflecting greenness of vegetation is a limitation in the sense that greenness does not directly reflect photosynthetic activity, e. g. a forest can be green for several weeks during water or heat stress, when photosynthesis is already diminishing. Thus NDVI is not the best proxy for vegetation activity at daily to weekly time scales, but rather reflects changes in phenology. We point to this in the following sentence in ll. 40-41.

2. In line 297, the authors suggest radiation as the main driver of NDVI in tropical regions. However, the interpretation of the source cited after the claim does not seem entirely accurate. Nemani (2003) says the tropical area "may have either a sustained dry season or nearly perpetual cloud cover that limits solar radiation", implying water availability and radiation may both be limiting factors in the tropics. However, the study only mentions radiation but not water availability to support an explanation for its results, which constitutes an inaccurate interpretation of the reference.I recommend rephrasing the sentence to accurately reflect the reference.

We will rephrase this sentence to: "In the tropics, radiation is proposed to be *one of* the main driver*s* of *plant productivity* …"

3. Page 9, figure 2a: No legends are given for land cover classes represented by the two types of red colors that are used in the figure.
In Figs. 2a+b the color scheme refers to oscillation classes, only in Fig. 2c to land cover classes. The two red colors represent SSS and SSA classes, as explained in both sub-figures.

4. Page 10, line 235: In other parts of the manuscript, it is being referred to as evergreen broadleaf forest (EBF), but in line 235 it is broadleaf evergreen forest. Replacing it with the acronym would do as this is not its first appearance.

The change will be implemented.

5. Page 10, line 244: This is essentially a repetition of what was discussed in line 235 without adding any new information.

In ll. 235ff we discuss the map in Fig. 3a as 'standalone' result, while in ll. 241ff we further discuss the comparison with other static classifications such as land cover types (Fig. 3b+c, Sup. Figs. 10-12).

Other than the above issues, the work is well done. I am confident that it will be accepted after these issues are properly addressed.

References

de Jong, R., Verbesselt, J., Schaepman, M. E., & de Bruin, S. (2011). Detection of breakpoints in global NDVI time series. De Jong, Rogier; Verbesselt, Jan; Schaepman, Michael E; de Bruin, Sytze (2011). Detection of Breakpoints in Global NDVI Time Series. In: 34th International Symposium on Remote Sensing of Environment (ISRSE), Sydney (AUS), 10 April 2011 -15 April 2011, Online., online. https://doi.org/info:doi/10.5167/uzh-77356Gitelson, A. A. (2004). Wide Dynamic Range Vegetation Index for Remote Quantification of Biophysical Characteristics of Vegetation. Journal of Plant Physiology, 161(2), 165–173. https://doi.org/10.1078/0176-1617-01176

Hawinkel, P., Swinnen, E., Lhermitte, S., Verbist, B., Van Orshoven, J., & Muys, B. (2015). A time series processing tool to extract climate-driven interannual vegetation dynamics using Ensemble Empirical Mode Decomposition (EEMD). Remote Sensing of Environment, 169, 375–389. https://doi.org/10.1016/j.rse.2015.08.024

Huang, M., Piao, S., Ciais, P., Peñuelas, J., Wang, X., Keenan, T. F., ... Janssens, I. A. (2019). Air temperature optima of vegetation productivity across global biomes. Nature Ecology & Evolution, 3(5), 772–779. https://doi.org/10.1038/s41559-019-0838-xHuete, A., Didan, K., Miura, T., Rodriguez, E. P., Gao, X., & Ferreira, L. G. (2002). Overview of the radiometric and biophysical performance of the MODIS vegetation indices. Remote Sensing of Environment, 83(1), 195–213. https://doi.org/10.1016/S0034-4257(02)00096-2

Huete, A. R. (1988). A soil-adjusted vegetation index (SAVI). Remote Sensing of Environment, 25(3), 295–309. https://doi.org/10.1016/0034-4257(88)90106-X

Nemani, R. R. (2003). Climate-Driven Increases in Global Terrestrial Net Primary Production from 1982 to 1999. Science, 300(5625), 1560–1563. https://doi.org/10.1126/science.1082750

Pettorelli, N., Vik, J. O., Mysterud, A., Gaillard, J.-M., Tucker, C. J., & Stenseth, N. Chr. (2005). Using the satellite-derived NDVI to assess ecological responses to environmental change. Trends in Ecology & Evolution, 20(9), 503–510. https://doi.org/10.1016/j.tree.2005.05.011

**References**

Braswell, B. H., Schimel, D. S., Linder, E., & Moore, B. (1997). The Response of Global Terrestrial Ecosystems to Interannual Temperature Variability. *Science*, *278*(5339), 870–873. https://doi.org/10.1126/science.278.5339.870

Braswell, Bobby H., Sacks, W. J., Linder, E., & Schimel, D. S. (2005). Estimating diurnal to annual ecosystem parameters by synthesis of a carbon flux model with eddy covariance net ecosystem exchange observations. *Global Change Biology*, *11*(2), 335–355. https://doi.org/10.1111/j.1365-2486.2005.00897.x

Frank, D., Reichstein, M., Bahn, M., Thonicke, K., Frank, D., Mahecha, M. D., … Zscheischler, J. (2015). Effects of climate extremes on the terrestrial carbon cycle: concepts, processes and potential future impacts. *Global Change Biology*, *21*(8), 2861–2880. https://doi.org/10.1111/gcb.12916

Hidayat, R. (2016). Modulation of Indonesian Rainfall Variability by the Madden-julian Oscillation. *Procedia Environmental Sciences*, *33*, 167–177. https://doi.org/10.1016/j.proenv.2016.03.067

Huete, A. (1997). A comparison of vegetation indices over a global set of TM images for EOS-MODIS. *Remote Sensing of Environment*, *59*(3), 440–451. https://doi.org/10.1016/S0034-4257(96)00112-5

Huete, A., Didan, K., Miura, T., Rodriguez, E. ., Gao, X., & Ferreira, L. . (2002). Overview of the radiometric and biophysical performance of the MODIS vegetation indices. *Remote Sensing of Environment*, *83*(1–2), 195–213. https://doi.org/10.1016/S0034-4257(02)00096-2

Kraft, B., Jung, M., Körner, M., Requena Mesa, C., Cortés, J., & Reichstein, M. (2019). Identifying Dynamic Memory Effects on Vegetation State Using Recurrent Neural Networks. *Frontiers in Big Data*, *2*. https://doi.org/10.3389/fdata.2019.00031

Krich, C., Runge, J., Miralles, D. G., Migliavacca, M., Perez-Priego, O., El-Madany, T., … Mahecha, M. D. (2019). Causal networks of biosphere-atmosphere interactions. *Biogeosciences Discussions*, 1–43. https://doi.org/10.5194/bg-2019-297

Madden, R. A., & Julian, P. R. (1971). Detection of a 40–50 Day Oscillation in the Zonal Wind in the Tropical Pacific. *Journal of the Atmospheric Sciences*, *28*(5), 702–708. https://doi.org/10.1175/1520-0469(1971)028<0702:DOADOI>2.0.CO;2

Mahecha, M. D., Gans, F., Brandt, G., Christiansen, R., Cornell, S. E., Fomferra, N., … Reichstein, M. (2019). Earth system data cubes unravel global multivariate dynamics. *Earth System Dynamics Discussions*, 1–51. https://doi.org/10.5194/esd-2019-62

Martínez, B., & Gilabert, M. A. (2009). Vegetation dynamics from NDVI time series analysis using the wavelet transform. *Remote Sensing of Environment*, *113*(9), 1823–1842. https://doi.org/10.1016/j.rse.2009.04.016

Mayta, V. C., Ambrizzi, T., Espinoza, J. C., & Dias, P. L. S. (2019). The role of the Madden–Julian oscillation on the Amazon Basin intraseasonal rainfall variability. *International Journal of Climatology*, *39*(1), 343–360. https://doi.org/10.1002/joc.5810

Nowosad, J., & Stepinski, T. F. (2018). Spatial association between regionalizations using the information-theoretical V-measure. *International Journal of Geographical Information Science*, *32*(12), 2386–2401. https://doi.org/10.1080/13658816.2018.1511794

Papagiannopoulou, C., Miralles, D. G., Decubber, S., Demuzere, M., Verhoest, N. E. C., Dorigo, W. A., & Waegeman, W. (2017). A non-linear Granger-causality framework to

investigate climate–vegetation dynamics. *Geoscientific Model Development*, *10*(5), 1945–1960. https://doi.org/10.5194/gmd-10-1945-2017

Pinzon, J., & Tucker, C. (2014). A Non-Stationary 1981–2012 AVHRR NDVI3g Time Series. *Remote Sensing*, *6*(8), 6929–6960. https://doi.org/10.3390/rs6086929

Vukićević, T., Braswell, B. H., & Schimel, D. (2001). A diagnostic study of temperature controls on global terrestrial carbon exchange. *Tellus B*, *53*(2), 150–170. https://doi.org/10.1034/j.1600-0889.2001.d01-13.x

Wang, B., Chen, G., & Liu, F. (2019). Diversity of the Madden-Julian Oscillation. *Science Advances*, *5*(7), eaax0220. https://doi.org/10.1126/sciadv.aax0220

Zhang, C. (2013). Madden–Julian Oscillation: Bridging Weather and Climate. *Bulletin of the American Meteorological Society*, *94*(12), 1849–1870. https://doi.org/10.1175/BAMS-D-12-00026.1

---

## Author Response (AR1)

Dear Akihiko Ito,

thank you for the positive evaluation of our response to the reviewers. We have incorporated the suggested changes into the manuscript. Please find attached a list of track changes, as well as the revised manuscript with highlighted track changes.

Kind regards,

The authors

**Track changes for: Towards a global understanding of vegetation–climate dynamics at multiple time scales**

Nora Linscheid[1,*], Lina M. Estupinan-Suarez[1,*], Alexander Brenning[2,3], Nuno Carvalhais[1,4], Felix Cremer[2,5], Fabian Gans[1], Anja Rammig[6], Markus Reichstein[1,3,7], Carlos A. Sierra[1], and Miguel D. Mahecha[1,7]

[*]These authors contributed equally.

[1]Max Planck Institute for Biogeochemistry, Hans–Knoell–Str. 10, 07745 Jena, Germany.

[2]Department of Geography, Friedrich Schiller University Jena, Loebdergraben 32, 07743 Jena, Germany.

[3]Michael Stifel Center Jena for Data–Driven & Simulation Science, Ernst–Abbe–Platz 2, 07743 Jena, Germany

[4]Departamento de Ciencias e Engenharia do Ambiente, DCEA, Faculdade de Ciencias e Tecnologia, FCT Universidade Nova de Lisboa, Caparica, Portugal

[5]Institute for Data Science, German Aerospace Center DLR, 07745 Jena, Germany

[6]TUM School of Life Sciences Weihenstephan, Technical University of Munich, Hans–Carl–von–Carlowitz–Platz 2, 85354 Freising, Germany.

[7] German Centre for Integrative Biodiversity Research (iDiv), Deutscher Platz 5e, 04103 Leipzig, Germany.

*Line numbers refer to the manuscript with track changes, attached after this document.*

**1 Introduction**

[revised manuscript text omitted]

Figure S5: **Assessment of NDVI GIMMS quality flags; direct observations and effect of retrieval values. a.** Median of fraction of direct observations at 0.5° per grid cell calculated overall time period (1982–2015). Fraction of direct observations ranges from 0 to 1, and corresponds to the number of pixels with direct observation after data aggregation (from 0.083° to 0.5°). Quality flag 1 is obtained when all aggregated pixels are direct observations, 0 if none are direct observations, **b.** Pixels that change NDVI dominant oscillation class when 0.3, 0.5, 0.7, 0.9, and 0.95 quality threshold is applied (quality is defined as the fraction of pixels originating from direct observations after aggregation), **c.** Percentage of pixels with change per dominant oscillation class. S: Short–term, A: Seasonal, L: Longer–term, T: Trend, in order from / to. Categories with change <0.05% are omitted. **d.** Median fraction originating from direct observation per pixel shown as box plot per oscillation regime. Lowest percentage of direct observation is found in seasonal NDVI regimes.

[Figure]

Figure S6: **Comparison of dominant oscillation classification between vegetation indices.** Dominant scale of variability for GIMMS NDVI from 1982 to 2015 (top), MODIS NDVI from 2001 to 2015 (center), and EVI MODIS from 2001 to 2015 (bottom). Dominant scale of variability was determined per pixel from normalized, detrended and Fourier–decomposed time series as the time scale containing highest relative variance.

[Figure]

Figure S7: **Assessment of the effect of land cover change over time on decomposition results of NDVI time series** Four pixels with >25% change in vegetation type according to Song et al. (2018) are displayed (columns), representing from left to right: (i) short vegetation gain, (ii) bare ground loss, (iii) bare ground gain, and (iv) tree loss. Rows from top to bottom: integrated NDVI signal (black), short–term oscillation (blue), seasonal oscillation (red), longer–term oscillation (green), and trend (yellow). Time series were normalized and detrended before Fourier decomposition.

[Figure]

Figure S8: **Bicolor map of undecomposed time series (top) and detrended, deseasonalized anomalies (bottom).** Pearson correlation of NDVI with precipitation (Prec, legend x axis) and air temperature ($T_{air}$, legend y axis) is shown at each grid cell. NDVI was lagged one time step (15 days) behind precipitation to allow response time, $T_{air}$ was correlated instantaneously. Color scale represents both correlations, binned into quantiles (e. g. purple – high positive correlation of NDVI with both $T_{air}$ and Prec, green – high negative correlation of NDVI with both $T_{air}$ and Prec). Data points where NDVI < 0.2 were excluded to avoid influence of inactive vegetation or non–vegetated time points.

[Figure]

Figure S9: **Bicolor map of Spearman correlations between NDVI, air temperature ($T_{air}$) and precipitation (Prec).** Correlation of NDVI with $T_{air}$ (legend y axis) and NDVI with Prec (legend x axis) were calculated between decomposed signals at each grid cell for each time scale (rows). NDVI was lagged one time step (15 days) behind precipitation to allow response time, $T_{air}$ was correlated instantaneously. Color scale represents both correlations, binned into quantiles (e. g. purple – high positive correlation of NDVI with both $T_{air}$ and Prec, green – high negative correlation of NDVI with both $T_{air}$ and Prec). Data points where NDVI < 0.2 were excluded to avoid influence of inactive vegetation or non–vegetated time points. The semi–annual cycle is included in the seasonal band.

[Figure]

Figure S10: **Bicolor map of Partial correlations between NDVI, air temperature ($T_{air}$) and precipitation (Prec).** Correlation of NDVI with $T_{air}$ (legend y axis) and NDVI with Prec (legend x axis) were calculated between decomposed signals at each grid cell for each time scale (rows). NDVI was lagged one time step (15 days) behind precipitation to allow response time, $T_{air}$ was correlated instantaneously. Color scale represents both correlations binned into quantiles (e. g. purple – high positive correlation of NDVI with both $T_{air}$ and Prec, green – high negative correlation of NDVI with both $T_{air}$ and Prec). Data points where NDVI < 0.2 were excluded to avoid influence of inactive vegetation or non–vegetated time points. The semi–annual cycle is included in the seasonal band.

[Figure]

Figure S11: **Bicolor map of Pearson correlations between MODIS EVI, air temperature ($\mathbf{T}_{air}$) and precipitation (Prec).** Correlation of EVI with $T_{air}$ (legend y axis) and MODIS EVI with Prec (legend x axis) were calculated between decomposed signals at each grid cell for each time scale (rows) for the years 2007–2015. EVI was lagged one time step (15 days) behind precipitation to allow response time, $T_{air}$ was correlated instantaneously. Color scale represents both correlations binned into quantiles (e. g. purple – high positive correlation of EVI with both $T_{air}$ and Prec, green – high negative correlation of with both $T_{air}$ and Prec). The semi–annual cycle is included in the seasonal band.

[revised manuscript text omitted]

*A: Annual, S: Short-term oscillation, L: Longer-term oscillation, T: Trend

Table S4: Spatial association between co–oscillation regimes and Köppen–Geiger or Global Land Cover (GLC2000), respectively. c = complementarity, h = homogeneity, m = number of classes , V = V–measure.

|  | Co-oscillations regime (11 classes) | | | |
| --- | --- | --- | --- | --- |
| **Static maps** | **m** | **h** | **c** | **V** |
| Köppen–Geiger | 4 | 0.19 | 0.16 | 0.17 |
| Global land cover | 9 | 0.16 | 0.09 | 0.11 |

**References**

Song, X.-P., Hansen, M. C., Stehman, S. V., Potapov, P. V., Tyukavina, A., Vermote, E. F., & Townshend, J. R. (2018, August). Global land change from 1982 to 2016. *Nature*, *560*, 639–643. doi: 10.1038/s41586-018-0411-9

---

## Author Response (AR2)

Dear Akihiko Ito,

thank you for accepting our manuscript for publication. We have incorporated the requested changes for the final version. We have further updated the 'Data availability' section with references to the supplementary notebook and data.

- The supplementary data and code are uploaded to Zenodo (zenodo.org) and will be made available under doi:10.5281/zenodo.3611262 upon publication with the complete reference to the published paper.
- Could we kindly ask you to add ORCIDs for Markus Reichstein (0000-0001-5736-1112) and Alexander Brenning (0000-0001-6640-679X) to the author list?

Many thanks in advance.

Sincerely,
The authors